# RESEARCHERBENCH:
# EVALUATING DEEP AI RESEARCH SYSTEMS ON THE FRONTIERS OF AI RESEARCH

## ABSTRACT

The emergence of deep research systems presents significant capabilities in problem-solving, extending from basic queries to sophisticated research tasks. However, existing benchmarks primarily evaluate these systems on web retrieval and report generation abilities, overlooking their potential for discovering, integrating and generating insights in AI research. To address this gap, we introduce ResearcherBench, the first benchmark focused on evaluating the capabilities of these advanced, agentic systems — which we refer to as Deep AI Research Systems (DARS) — on frontier AI research questions. We curated a dataset of 65 research questions expertly selected from real-world AI research scenarios such as laboratory discussions and interviews, spanning 35 different AI subjects and categorized into three types: technical details, literature review, and open consulting. Our dual evaluation framework combines rubric assessment, which uses expert-designed criteria to evaluate insight quality, with factual assessment, which measures citation accuracy (faithfulness) and coverage (groundedness). We evaluated several leading commercial DARS and baseline systems. Our evaluation results reveal the strengths and limitations of these systems, with particular strength in open-ended consulting questions compared to technical implementation tasks. Such capabilities demonstrate the potential for DARS to serve as genuine AI research partners, representing a meaningful step toward AI self-improvement. We open-source ResearcherBench to provide a standardized platform for promoting the development of next-generation AI research assistants, hoping to foster a new perspective in AI research evaluation for scientific collaboration.

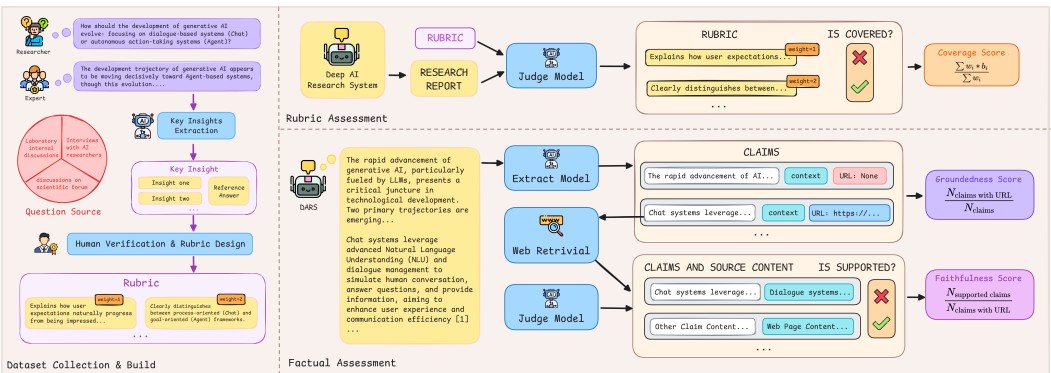

Figure 1: ResearcherBench Framework Overview. The framework consists of three main components: (1) Dataset collection from authentic research scenarios leading to expert-generated rubrics, (2) Rubric assessment to evaluate coverage against rubrics, and (3) Factual assessment to measure faithfulness and groundedness scores.

# 1 INTRODUCTION

The advent of artificial intelligence has fundamentally transformed how we approach complex problem-solving tasks, with deep research systems emerging as sophisticated AI agents capable of autonomously conducting intricate research workflows (Gridach et al., 2025; Huang et al., 2025). These systems demonstrate remarkable proficiency in information retrieval, analysis, and synthesis across diverse domains (Xu & Peng, 2025), with increasing deployment in AI Research for technical investigation, literature review, and knowledge synthesis (Stanford HAI, 2025).

These applications have demonstrated clear value in streamlining traditional research workflows and enhancing productivity in well-established AI research domains. However, the community might overlook a potentially transformative capability: their potential to assist researchers in exploring genuinely open-ended, frontier AI research questions that exist at the cutting edge of artificial intelligence knowledge (Lu et al., 2024; Research, 2025).

The transition from being systems for information retrieval and summarization to becoming genuine "research partners" capable of valuable collaboration on unexplored AI Research territories represents a significant challenge facing the development of AI research assistants today (Wang & Chen, 2024). To address this challenge, we define this emerging category as **Deep AI Research Systems (DARS)**: sophisticated agentic systems that combine dynamic reasoning capabilities with autonomous research workflow execution, including multi-iteration web retrieval, tool utilization, and iterative refinement processes (Huang et al., 2025).

This capability gap raises a critical question: can current DARS truly assist human researchers in tackling the most challenging, high-valued, and open-ended questions at the frontiers of AI research, where definitive answers do not yet exist and novel insights must be synthesized from fragmentary and cross-domain information within the artificial intelligence field?

Answering this question demands fundamentally different evaluation frameworks. However, existing benchmarks for evaluating DARS capabilities predominantly focus on comprehensive report generation abilities (Du et al., 2025) or web interaction capabilities (Gou et al., 2025; Wei et al., 2025), but fail to assess the crucial ability to understand, analyze, and provide meaningful insights on highly specialized, cutting-edge AI research problems (Zheng et al., 2025; Starace et al., 2025).

To address this gap, we introduce **ResearcherBench**, the first benchmark specifically designed to evaluate DARS capabilities on frontier AI research questions, as shown in Figure 1. We initially collected an initial corpus of 932 AI research questions from authentic AI research scenarios, and filtered them to a curated dataset of 65 high-quality questions spanning 35 different AI subjects and categorized into three types. We also developed a dual evaluation framework combining human expert-designed rubric assessment with factual assessment, to evaluate both the intellectual depth and factual reliability of DARS responses.

ResearcherBench complements existing evaluation methods by focusing on whether DARS can provide meaningful assistance to human researchers working on genuinely unsolved, cutting-edge problems in the field of artificial intelligence. Our benchmark represents a paradigm shift in evaluation philosophy—moving from assessing "whether systems can retrieve and summarize information" to evaluating "whether DARS can understand complex problems and provide meaningful insights as genuine research partners." Our main contributions are stated as follows:

- **A Novel Task Collection:** We present a carefully expert-reviewed and curated dataset of 65 high-quality AI research questions sourced from authentic frontier scenarios, spanning 35 distinct AI subjects and being categorized into three distinct types.

- **A Dual Evaluation Framework:** Our assessment methodology combines human expert-designed rubric assessment with factual assessment, evaluating both insight quality and citation reliability.

- **Comprehensive Empirical Analysis:** Our evaluation of leading commercial DARS reveals capability and limitations, including superior performance on open-ended tasks over technical synthesis and a paradoxical "high faithfulness, low groundedness" pattern.

- **Open-Source Contribution:** We publicly release ResearcherBench as a comprehensive evaluation platform, encompassing our curated dataset and evaluation framework to collaboratively foster the development of AI research assistance capabilities.

| Benchmark | Evaluation Focus | Requires Real-time Web | Open-ended Research Query | Human Expert-designed Rubric |
|---|---|---|---|---|
| ResearchBench (Liu et al., 2025) | Hypothesis Generation | ✗ | ✓ | ✗ (Automated metrics) |
| Mind2Web 2 (Gou et al., 2025) | Web Infomation Retrieval | ✓ | ✗ | ✗ (LLM-generated) |
| DeepResearch Bench (Du et al., 2025) | General Research Tasks | ✓ | ✗ | ✗ (LLM-generated) |
| **ResearcherBench (Ours)** | **Frontier Research Tasks** | ✓ | ✓ | ✓ |

Table 1: Comparison of Deep Research-related Benchmarks

## 2 RELATED WORK

### 2.1 DEEP AI RESEARCH SYSTEMS

**Early DARS.** Early Deep AI Research Systems (DARS) evolved from basic prompt-based approaches (Zheng et al., 2024; Alzubi et al., 2025) and supervised fine-tuning methods (Asai et al., 2024), built upon retrieval-augmented generation (RAG) architectures within static knowledge repositories. These systems faced limitations including reliance on non-updatable knowledge bases and lack of iterative reasoning for complex research workflows.

**Modern DARS.** Modern DARS address these limitations through reinforcement learning-based approaches (Zheng et al., 2025; Jin et al., 2025; Chen et al., 2025; Song et al., 2025), implementing multi-agent architectures for autonomous web search with iterative query formulation and dynamic refinement. Commercial implementations of Deep Research System (OpenAI, 2025; Google, 2025; xAI, 2025; Anthropic, 2025) demonstrate these capabilities, generating high-quality research reports through iterative workflows.

### 2.2 RESEARCH-RELATED EVALUATION BENCHMARKS

**Traditional RAG-based Benchmarks.** Early evaluation frameworks established multi-hop reasoning assessment requiring information synthesis across sources (Joshi et al., 2017; Yang et al., 2018). OpenScholar (Asai et al., 2024) evaluated literature synthesis with expert-written responses, emphasizing citation accuracy across AI research domains. These benchmarks focused on factual retrieval and basic reasoning.

**Specialized Research Task Benchmarks.** Recent benchmarks target comprehensive research evaluation. ResearchBench (Liu et al., 2025) evaluates LLMs' scientific hypotheses discovery capabilities, while DeepResearch Bench (Du et al., 2025) introduces RACE and FACT methodologies to assess Deep Research Agents' report generation quality and information retrieval and grounding capabilities. PaperBench (Starace et al., 2025) and EXP-Bench (Kon et al., 2025) evaluate research replication capabilities on gradable academic tasks. These benchmarks shift focus toward established technical competencies and knowledge synthesis.

**Agentic Search and Browsing Benchmarks.** Parallel developments assess agentic search capabilities. Mind2Web 2 (Gou et al., 2025) introduces 130 long-horizon tasks requiring real-time web browsing, implementing Agent-as-a-Judge framework for automated assessment. BrowseComp (Wei et al., 2025) measures persistent browsing abilities through navigation tasks. These benchmarks prioritize information retrieval breadth and web interaction capabilities.

**Limitations.** Despite these advances, existing benchmarks focus on established competencies like report generation and web interaction, prioritizing information retrieval over the conceptual understanding and novel insight generation required for frontier AI research. Moreover, evaluation rubrics for frontier AI research problems cannot be reliably generated by automated systems, requiring extensive AI domain expertise to assess valuable insights in cutting-edge AI research. Table 1 presents a comparison of several related benchmarks.

This gap motivates the need for evaluation frameworks that can assess AI systems' potential as genuine research partners in exploring uncharted AI research territories, where novel insight generation and deep conceptual understanding of AI problems are paramount.

## 3   RESEARCHERBENCH

ResearcherBench presents a systematic approach to constructing a comprehensive benchmark for evaluating DARS capabilities on frontier AI research questions. We employed rigorous data collection and filtering methodologies to curate AI research questions from real-world scientific scenarios, resulting in a high-quality dataset of 65 questions across 35 subjects.

### 3.1   DATA COLLECTION AND CATEGORIZATION

We collected frontier AI research questions from authentic scientific scenarios, resulting in an initial corpus of 932 candidate questions. These questions are from three primary sources: laboratory research discussions, interviews with leading AI researchers, and scientific forum discussions. Questions were deemed "frontier AI research questions" if they target core, unsolved limitations of current AI domains, requiring novel insights to address fundamental challenges (Marcus, 2020).

We utilized `Claude-3.7-Sonnet` for automated classification of our questions across different types and AI subjects, followed by manual review to ensure accuracy. we categorized questions into three types based on their cognitive demands: Technical Details, Literature Review, and Open Consulting. The subject-based classification yielded 35 distinct AI research areas including model architecture, multimodal fusion, AI ethics, and other emerging paradigms. It ensures a comprehensive coverage of contemporary AI research challenges. Details in data collection, definition of question types and graph of topic distribution are provided in Appendices B.1, B.2 and B.3.

### 3.2   QUESTION SELECTION AND FILTERING

We developed a **type-specific selection framework** to filter high-quality AI research questions, recognizing that different question categories require distinct quality criteria. Instead of uniform evaluation standards, we designed tailored evaluation criteria for each question type: technical details questions emphasized methodological precision and verifiability, literature review questions focused on comprehensive scope and consistency, while open consulting questions prioritized conceptual depth and innovative perspectives.

Each question received independent evaluation from at least two experienced researchers with domain expertise in the corresponding AI field. Through this systematic review process, we refined the initial corpus from 932 to 65 high-quality questions that met our stringent selection standards. This selective framework prioritizes quality over dataset size, aligning with our goal of assessing DARS capabilities on genuinely frontier and challenging AI research scenarios. Detailed evaluation methodology, criteria and examples are provided in Appendices B.4 and B.5.

## 4   DUAL EVALUATION FRAMEWORK

ResearcherBench introduces a dual evaluation framework that comprehensively assesses DARS performance through both **rubric assessment** (evaluating insight quality and conceptual depth) and **factual assessment** (measuring citation accuracy and content groundedness). This complementary approach captures both the intellectual contribution and empirical reliability of research responses.

### 4.1   RUBRIC ASSESSMENT

Traditional LLM-as-a-Judge evaluation (Gu et al., 2025; Li et al., 2025) with single overall ratings provides insufficient granularity for assessing complex research responses. Frontier research evaluation requires nuanced assessment across multiple dimensions that single scores cannot capture.

Our fine-grained rubric assessment framework addresses this limitation by decomposing complex research questions into multiple specific, expert-designed criteria with assigned importance weights. This approach enables systematic evaluation of research insight quality across conceptual understanding, methodological rigor, and analytical depth. This structured approach provides the granular assessment necessary for evaluating the quality of complex research insights.

### 4.1.1 RUBRIC CONSTRUCTION

Our evaluation rubrics are designed by experts in various AI domains to reflect the standards and expectations of frontier AI research. We employ a three-stage method to construct these rubrics:

**Stage 1: Insight Extraction.** `Claude-3.7-Sonnet` is employed to analyze and extract key insights from multiple diverse contextual sources for each question, including original discussion records, expert opinions, and technical background materials, and industry practice cases. This integration process generates comprehensive reference materials that contain these high-value insights. Domain experts subsequently review, validate, and supplement these integrated materials to ensure accuracy and completeness, establishing a solid foundation for subsequent rubric development.

**Stage 2: Criteria Design.** Following standardized rubric design guidelines and templates, we invited human annotators (experienced masters, Ph.D.s or professionals specialized in AI) to design evaluation rubrics with multiple criteria independently. The reference materials with key insights extracted in the Stage 1 serve as auxiliary information — annotators used them to ensure comprehensive coverage of rubric. Each criterion represents a specific aspect of reasoning or insight that should be addressed in a comprehensive response, assigned with an importance weight (1-3 scale) based on their relative importance on overall answer quality. Detailed guidelines for rubric construction and examples are provided in Appendix C.

**Stage 3: Quality Control.** To ensure rubric reliability and validity, we implemented a multi-stage quality control process. Each rubric was collaboratively developed by 2-3 experienced AI researchers, with one annotator responsible for initial drafting and weight assignment, and the others conducting comprehensive review and collaborative refinement. All rubrics underwent expert review to assess completeness, clarity, independence, and discriminative power. We conducted pilot testing with 20 sample DARS responses to identify and refine problematic criteria through iterative revision. Finally, we validated rubric effectiveness through meta-evaluation comparing human expert judgments with automated assessment, as detailed in Section 4.3.

### 4.1.2 EVALUATION METRIC

We define Coverage Score $S_c$ as a weighted metric of how comprehensively a DARS response cover the key insights specified in the expert-designed rubric. Let $Q_k$ denote the $k$-th question in our benchmark, and $\text{Res} = \text{DARS}(Q_k)$ represent the final response generated by DARS. The binary indicator $b_i \in \{0, 1\}$ for each criterion is computed as:

$$b_i = \text{Judge}(Q_k, \text{Res}, r_i), \tag{1}$$

where $r_i$ represents the content of the $i$-th criterion, and $\text{Judge}(\cdot)$ returns 1 if the response satisfies criterion $r_i$, and 0 otherwise. For each question $Q_k$, the Coverage Score is calculated as:

$$S_c = \frac{\sum_{i=1}^{n} w_i \cdot b_i}{\sum_{i=1}^{n} w_i}, \tag{2}$$

where $w_i \in \{1, 2, 3\}$ represents the weight of the $i$-th criterion, and $n$ is the total number of criteria. This formulation reflects both the coverage of expert-aligned criteria and the relative importance across different criteria.

## 4.2 FACTUAL ASSESSMENT

While rubric assessment evaluates insight quality and conceptual depth, factuality remains a fundamental requirement for reliable DARS-generated research reports (Zhang et al., 2023). Our factual assessment framework addresses the challenges of evaluating faithfulness and groundedness (ExplodingGradients, 2024) in automated research synthesis.

### 4.2.1 ASSESSMENT METHODOLOGY

**Stage 1: Claim Extraction.** We employ an extraction model to identify all factual claims within DARS-generated reports along with their contextual information. For each extracted claim, the

model determines whether it corresponds to a citation URL in the report's reference section. Claims with valid citations are saved as URL-claim-context triplets for subsequent verification, Otherwise, this claim is considered ungrounded, and its URL is marked as empty.

For question $Q_k$, let $\text{Res}, \text{Ref} = \text{DARS}(Q_k)$ represent the main content and reference section of the research report generated by DARS. We define a claim $c_i$ as a triplet $(\text{text}_i, \text{context}_i, \text{url}_i)$, where $\text{text}_i$ is the textual content of the claim extracted from responses, $\text{url}_i \in \text{URL} \cup \{\emptyset\}$ is the associated citation URL extracted from references (if exists), and $\text{context}_i$ is the context surrounding the claim used as supplementary information for verification. The complete set of extracted claims $C_k$ for question $Q_k$ is then represented as:

$$C_k = \{c_i \mid c_i = \text{Extract}(Q_k, \text{Res}, \text{Ref}), i = 1, 2, ..., N_k\}, \tag{3}$$

where $\text{Extract}(Q_k, \text{Res}, \text{Ref})$ is the extraction model that identifies factual statements in the Res, and $N_k = |C_k|$ is the total number of claims for $Q_k$.

**Stage 2: Citation Support Verification.** For each claim, we extract the entire textual contents from URL sources using the Jina Reader API[1]. Then a judge model performs binary verification to determine whether the extracted content supports the corresponding claim. When claims are semantically incomplete or ambiguous in isolation, the surrounding context serves as supplementary information to assist the model's judgment.

Denoted that $C_k^{cited} \subseteq C_k$ is the subset of cited claims with non-empty URLs, and $C_k^{supp} \subseteq C_k^{cited}$ is the subset of claims that are both cited and supported by their URL sources, we define:

$$C_k^{cited} = \{c_i = (\text{text}_i, \text{context}_i, \text{url}_i) \mid c_i \in C_k \text{ and } url_i \neq \emptyset\},$$
$$C_k^{supp} = \{c_i \in C_k^{cited} \mid \text{Judge}(\text{text}_i, \text{context}_i, \text{SourceText}(\text{url}_i)) = 1\},$$

where $\text{SourceText}(\text{url})$ represents the entire textual content extracted from the URL, and $\text{Judge}(\text{text}, \text{context}, \text{SourceText}(\text{url}))$ returns 1 if the claim is supported by the URL, 0 otherwise.

### 4.2.2 EVALUATION METRIC

Based on verification results, two complementary metrics are calculated to assess the overall factual reliability of DARS-generated report:

- **Faithfulness Score** $S_f$, which measures the accuracy of citations in supporting their corresponding claims. This metric evaluates the proportion of cited claims that are actually supported by their referenced sources, indicating the reliability of citation-claim relationships when citations are provided.
- **Groundedness Score** $S_g$, which measures the overall citation coverage of response content. This metric evaluates the proportion of all factual claims that have explicit citation support, reflecting how comprehensively the research report grounds its assertions in verifiable sources rather than relying on unsupported statements.

Suppose that $N_{c,k} = |C_k^{cited}|$ is the total number of cited claims, and $N_{s,k} = |C_k^{supp}|$ is the total number of supported claims, these two metric are calculated as:

$$S_f = \frac{N_{s,k}}{N_{c,k}} \ , \ \ S_g = \frac{N_{c,k}}{N_k} \ . \tag{4}$$

### 4.3 JUDGE MODEL SELECTION VIA HUMAN EVALUATION

Selecting an appropriate automated judge model is critical for benchmark evaluation. We address this through systematic human expert evaluation: by comparing multiple candidate LLM judges against ground-truth human expert assessments, we can identify which automated judge achieves optimal alignment with expert-level evaluation standards.

This selection process ensures that our automated scoring mechanism can reliably reflect human expert judgment, while enabling comprehensive and scalable evaluation across our full benchmark.

---

[1] https://jina.ai/reader

| Judge Model | Acc. | Prec. | Rec. | F1 | Cost |
|---|---|---|---|---|---|
| DeepSeek R1 | 0.71 | 0.83 | 0.68 | 0.75 | 0.23 |
| Gemini 2.5 Flash | 0.72 | 0.75 | 0.82 | 0.78 | 0.54 |
| GPT-4.1 | 0.72 | 0.75 | **0.85** | 0.80 | 0.19 |
| o3 | **0.76** | 0.80 | 0.83 | **0.81** | 0.22 |
| o3-mini | **0.76** | **0.85** | 0.76 | 0.80 | **0.13** |

Table 2: Performance comparison of candidate judge models against human annotations. Cost represents average API expense per question evaluation ($). Bold values indicate best performance per metric. The evaluation considers the assigned importance for each criterion in the rubric assessment.

### 4.3.1 EXPERIMENTAL DESIGN

**Human Expert Annotation for Ground Truth.** To establish reliable reference standards for judge model selection, we firstly conducted human expert evaluation. Given the intensive nature of expert evaluation (1.5 hours per response for rubric assessment), we carefully constructed a stratified validation subset of 20 responses spanning different question types, difficulty levels, and DARS systems to ensure representativeness. We recruited a team of 5 domain expert annotators to independently evaluate these responses using identical rubrics employed in our full benchmark assessment. Each response received evaluation from at least two independent annotators to ensure reliability. This process yielded reliable reference standards for judge model selection.

**Candidate Judge Models and Selection Criteria.** We systematically evaluated five leading LLMs as candidate automated judges: DeepSeek R1, Gemini-2.5-flash, GPT-4.1, o3, and o3-mini. Our selection methodology considered two complementary dimensions: (1) **Performance consistency**: measured by accuracy, precision, recall, and F1-score quantifying agreement between automated judgments and human expert annotations; and (2) **Cost efficiency**: API costs per question, enabling practical large-scale deployment. This dual-criteria approach ensures we select judge models that achieve both high-quality human alignment and practical scalability for comprehensive benchmark evaluation.

### 4.3.2 VALIDATION RESULTS AND FINAL SELECTION

Experiment shows that our Inter-annotator agreement (Cohen's Kappa) acieved $\kappa = 0.67$, indicating substantial consensus among independent experts. As shown in Table 2, top-performing models (`o3`, `o3-mini`, `GPT-4.1`) achieved F1-scores of 0.80-0.81, demonstrating substantial agreement with human experts (Landis & Koch, 1977).

Based on results above, we selected `o3-mini` for rubric assessment, due to the optimal trade-off between human alignment (F1: 0.80, precision: 0.85) and cost efficiency ($0.13 per question). For factual assessment, we chose `GPT-4.1` for its superior performance in long-context processing required for claim extraction and citation verification.

## 5 EXPERIMENTS AND RESULTS

### 5.1 EXPERIMENTAL SETUP

**Evaluated Systems.** We evaluated several leading commercial DARS to assess their performance on frontier AI research questions: OpenAI Deep Research (OpenAI, 2025), Gemini Deep Research powered by Gemini-2.5-Pro (Google, 2025), Claude Research (Anthropic, 2025), Grok DeepSearch & DeeperSearch (xAI, 2025), and Perplexity Deep Research (Perplexity AI, 2025). We also tested two commercial LLM platforms with Deep Research Mode: Doubao[2] and Mita[3]. To provide comprehensive comparison, we evaluated two LLM systems with web search tools: GPT-4o Search Preview and Perplexity Sonar Reasoning Pro. Details on our specific interaction procedures with each system can be found in Appendix D.

---

[2] https://www.doubao.com
[3] https://metaso.cn

**Evaluation Configuration.** After comprehensive evaluation of multiple candidate Judge LLMs (detailed in Section 4.3), we selected `o3-mini` as the judge model for rubric assessment to evaluate criteria coverage. For factual assessment, we chose `GPT-4.1` as both the extraction model and judge model. All evaluations were conducted between March and July in 2025 to ensure temporal consistency and fair comparison across systems. Details on data collection, specific implementation (e.g. prompt template), pairwise comparisons and error analysis are provided in Appendix E.

## 5.2 MAIN RESULTS

| Model | Rubric Assessment | Factual Assessment | |
| --- | --- | --- | --- |
| | Coverage | Faithfulness | Groundedness |
| *Deep AI Research Systems* | | | |
| OpenAI Deep Research | **0.7032**, [-0.050, +0.049] | 0.84 | 0.34 |
| Gemini Deep Research | 0.6929, [-0.053, +0.052] | **0.86** | 0.59 |
| Claude Research [†] | 0.6113, [-0.053, +0.053] | - | - |
| Mita Deep Research | 0.5835, [-0.057, +0.059] | 0.65 | 0.59 |
| Doubao Deep Research | 0.5754, [-0.057, +0.055] | 0.78 | 0.63 |
| Perplexity Deep Research | 0.4800, [-0.056, +0.055] | 0.85 | 0.56 |
| Grok3 DeepSearch | 0.4414, [-0.054, +0.057] | 0.69 | 0.32 |
| Grok3 DeeperSearch | 0.4398, [-0.056, +0.056] | 0.80 | 0.31 |
| *LLM with Search Tools* | | | |
| GPT-4o Search Preview | 0.3576, [-0.051, +0.052] | **0.86** | 0.39 |
| Perplexity: Sonar Reasoning Pro | 0.4663, [-0.056, +0.056] | 0.62 | **0.68** |

Table 3: Comprehensive evaluation results. Bold values indicate the best performance. We also report 95% bootstrap confidence intervals for Rubric Assessment Coverage using 10,000 bootstrap samples. † Factual assessment for Claude Research is unavailable due to limited citation extraction.

### 5.2.1 RUBRIC ASSESSMENT RESULTS

As shown in Table 3, OpenAI Deep Research and Gemini Deep Research achieve the highest coverage scores (0.70 and 0.69). Claude Research, Mita Deep Research, and Doubao Deep Research demonstrate mid-tier performance (0.58-0.61), while Perplexity Deep Research (0.48) and both Grok variants (0.44) achieving limited coverage. LLM with Search tools generally underperform dedicated DARS across all evaluated metrics, while Perplexity Sonar Reasoning Pro (0.46) outperforms two Grok Deep Research variants (0.44). The bootstrap analysis validates the robustness of our rankings, confirming that the rubric effectively distinguishes between different tiers.

Rubric assessment results reveal that: **Iterative web search combined with deep reasoning capability determines high quality in the rubric assessment.** Dedicated DARS demonstrate overall superior performance compared to LLM+Search tools (e.g., OpenAI Deep Research v.s. GPT-4o Search Preview), indicating that such models lack iterative search refinement and deep reasoning capacities, thus failing to address frontier AI Research questions. Notably, while Perplexity Sonar Reasoning Pro outperforms Grok Deep Research despite being a non-DARS system, it remains inferior to other dedicated DARS like Perplexity Deep Research, demonstrating that advanced reasoning capabilities can partially bridge the performance gap, but cannot fully substitute for iterative search mechanisms with deep reasoning, which is essential for comprehensive research synthesis.

### 5.2.2 FACTUAL ASSESSMENT RESULTS

The factual evaluation reveals a consistent "high faithfulness, low groundedness" pattern across most systems. As shown in Table 3, most DARS achieve strong faithfulness scores (0.65-0.86), indicating robust citation-claim relationships. However, groundedness scores vary substantially (0.31-0.68), with limited citation coverage observed across most systems. Despite being non-DARS systems, GPT-4o Search Preview reaches the highest faithfulness score, and Perplexity Sonar Reasoning Pro demonstrates the highest overall groundedness (0.68).

Factual assessment results reveal that: **Groundedness serves as a diagnostic indicator of DARS operational modes rather than a direct measure of research quality.** The observed variation in groundedness scores across DARS reveals fundamental differences in their operational approaches: high groundedness systems function primarily as "information aggregators", directly citing and summarizing source materials, while low groundedness systems operate as "analytical synthesizers", transforming retrieved information through inferential reasoning processes. This reframes groundedness from a quality metric to an operational indicator, distinguishing between systems that excel at comprehensive source compilation versus those optimized for cross-source knowledge integration and novel insight generation. Our case study analysis in Appendix F illustrates how these different operational modes produce distinct value of research outputs.

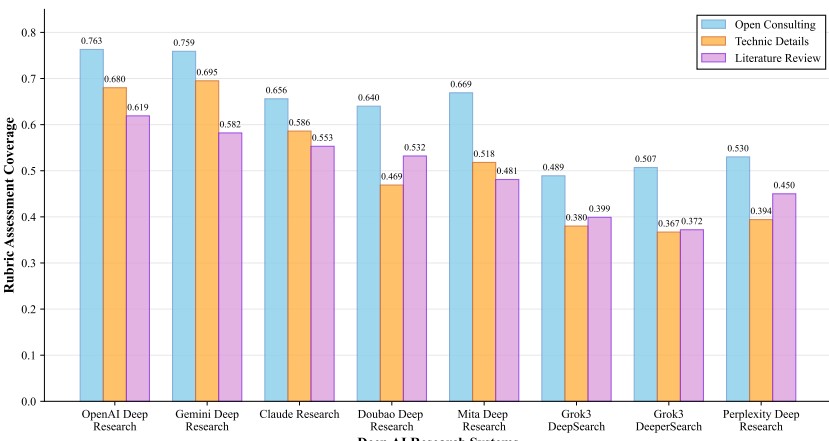

Figure 2: Performance comparison across different question types for DARS.

### 5.2.3 ANALYSIS ACROSS QUESTION TYPES

Analysis across question types reveals distinct capability patterns, as illustrated in Figure 2. All systems demonstrate superior performance on open-ended consulting questions than literature review tasks and technical detail questions, with top performers achieving coverage scores above 0.76. Results reveal that **Architectural design of DARS favors divergent exploration over precision retrieval.** Open consulting questions benefit from the divergent exploration and creative synthesis that DARS excel at, allowing them to generate novel insights through broad information integration. However, literature review tasks require precise retrieval of specific articles and comprehensive coverage of established work, while technical detail questions demand exact technical solutions and implementation specifics. The performance differences reveal that current DARS are most effective in exploratory synthesis, positioning them as research ideation tools rather than comprehensive literature surveyors or technical implementation guides.

## 6 CONCLUSION

This paper introduces ResearcherBench, the first benchmark for evaluating Deep AI Research Systems (DARS) on frontier AI Research questions, comprising 65 curated questions with dual rubric and factual assessment frameworks. Our comprehensive evaluation of leading commercial DARS establishes crucial performance baselines and reveals three key insights into automated research assistance capabilities. By open-sourcing ResearcherBench, we hope to promote a paradigm shift in AI evaluation from information retrieval assessment to genuine research partnership capability, providing the foundation for developing AI systems capable of meaningful AI Research collaboration. limitations and future directions for extending this work are discussed in Appendix A.

## REPRODUCIBILITY AND ETHICS STATEMENT

**Reproducibility:** To ensure the reproducibility of our findings, we provide a comprehensive set of resources. The complete source code and dataset are available in the supplementary material, and all implementation details are documented in the Appendix.

**Ethics:** We adhere to ICLR's ethical guidelines and have ensured compliance with all relevant legal and ethical standards. There was no involvement of human research subjects requiring IRB approval. All annotators participated voluntarily with their privacy and ethical rights fully protected, reasonable workload, and fair compensation. We declare no conflicts of interest or undisclosed funding sources. Our code and dataset contain no harmful content, and we have followed all ethical guidelines related to privacy, security, and research integrity.

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

# A LIMITATIONS AND FUTURE WORKS

## A.1 LIMITATIONS

While ResearcherBench represents a significant advance in evaluating DARS capabilities on frontier research questions, several limitations should be acknowledged.

**Domain Specificity.** Our benchmark focuses exclusively on AI-related research questions, which may limit the generalizability of findings to other scientific domains such as biology, physics, or chemistry. The specialized nature of AI research questions may exhibit different characteristics compared to frontier problems in other fields, and DARS performance patterns might vary across disciplines.

**Dataset Scale.** Our dataset size of 65 questions, while carefully curated for quality, represents a relatively small sample that may not capture the full spectrum of frontier research challenges. The distribution across question types (12 technical details, 20 literature review, 33 open consulting) reflects our current understanding of research assistance needs but may not represent the optimal balance for comprehensive evaluation.

| Question Type | Definition | Example |
|---|---|---|
| Technical Details | Questions requiring explanations of methodologies, implementations, or theoretical concepts with a strong emphasis on accuracy and verification. | `Why doesn't ChatGPT directly fine-tune using Reward-Model data, but instead use RLHF? Give me a more deep technical report, and focus on references to recent research papers on this topic.` |
| Literature Review | Questions that involve synthesizing findings from multiple research papers, comparing methodologies, and identifying trends or gaps in existing literature. | `For complex reasoning tasks, what are the strengths of current agent technologies, and what are their limitations? Please analyze this in the context of research since June 2024.` |
| Open Consulting | Questions that explore emerging trends, strategic insights, detailed solutions or broader implications, often requiring subjective interpretation and expert judgment. | `Could transformer architectures be fundamentally reimagined to process multimodal inputs (e.g. video, audio, or text) with the same efficiency they process text?` |

Table 4: Example of Different Question Types. Each question type follows a detailed definition and an example. Technical detail questions emphasize precise verification, literature review questions focus on comprehensive synthesis, and open consulting questions prioritize broader insights.

**Black-box Commercial Systems.** Our focus on commercial DARS systems limits insights into the fundamental architectural and training approaches that drive performance differences. The black-box nature of these systems prevents deeper analysis of why certain systems excel in specific question types or what design principles enable superior frontier research assistance.

A.2 FUTURE WORKS

Several promising directions emerge from our findings that warrant further investigation.

**Cross-Domain Expansion.** Expanding ResearcherBench to additional scientific domains would provide valuable insights into the domain-generalizability of DARS capabilities. Developing comparable benchmarks for fields such as biology, chemistry, physics, and social sciences would enable cross-domain analysis of research assistance patterns and reveal whether the open consulting superiority we observed generalizes beyond AI research.

**Continuous Benchmark Evolution.** As frontier research rapidly evolves, maintaining the relevance and currency of our benchmark requires periodic incorporation of new research questions that reflect the latest developments in AI and related fields. This ongoing evolution will ensure that ResearcherBench remains aligned with the cutting-edge of scientific inquiry, capturing emerging research paradigms and novel challenges that push the boundaries of current knowledge. Regular updates will involve collaboration with active researchers to identify and validate the most pressing and innovative questions in contemporary AI research.

**Longitudinal DARS Evaluation.** Implementing systematic longitudinal evaluation of DARS systems would provide crucial insights into capability development trajectories and technological advancement patterns. By continuously assessing new DARS releases and iterations on our benchmark, we can track performance changes over time, identify which aspects of frontier research assistance improve most rapidly, and analyze the developmental pathways of different system architectures. This longitudinal analysis would inform both research priorities and development strate-

gies, while providing the community with empirical evidence of progress in AI research assistance capabilities.

These future directions would collectively advance our understanding of AI research assistance capabilities and contribute to the development of systems that can serve as genuine partners in scientific discovery across diverse domains.

## B    DATASET CONSTRUCTION

### B.1    DETAILS IN DATA COLLECTION

The initial corpus of 932 candidate questions are from three primary sources: (1) **Laboratory research discussions (255 questions)**: We collected these questions from transcripts of laboratory discussions across various research topics, where researchers actively grapple with unsolved technical challenges; (2) **Interviews with leading AI researchers (344 questions)**: We gathered these questions from the subtitles of frontier interviews conducted on video platforms (such as YouTube), which often reveal emerging research directions; and (3) **Scientific forum discussions (333 questions)**: These questions were collected from research-oriented topics on online forums like X (formerly Twitter) and Zhihu, where researchers share their latest work and insights.

Data collection was conducted between October 2024 to April 2025, ensuring currency in the rapidly evolving AI field. We gathered the contextual information of each question, which serves as supporting material for the rubric construction process.

### B.2    DEFINITION OF QUESTION TYPES

Table 4 shows detailed definition and example of each question type.

### B.3    TOPIC DISTRIBUTION GRAPH

As shown in Figure 3, through subject-based classification, we identified 35 distinct AI research areas encompassing model architecture, multimodal fusion, AI ethics and other emerging paradigms, It achieve a comprehensive representation of contemporary AI research challenges.

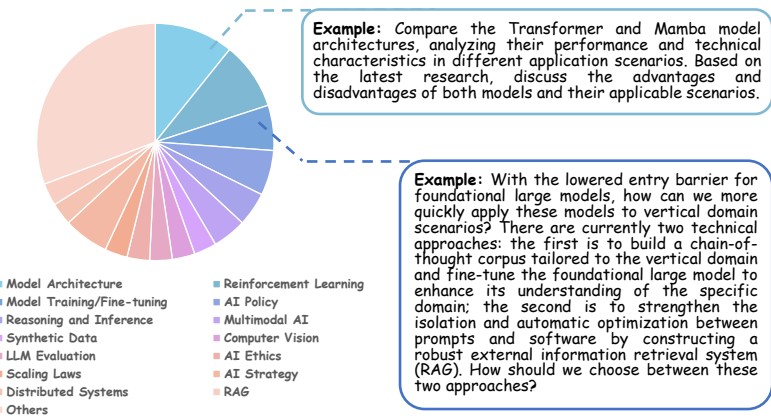

Figure 3: Topic Distribution of Benchmark. Left Side: Pie chart showing the distribution of AI subjects in the benchmark. Right Side: Question examples from major subjects.

### B.4    DETAILS IN QUESTION SELECTION

We formalized the tailored criteria of different question types into structured rubrics using 5-point Likert scales (1=Poor, 5=Excellent) (Joshi et al., 2015). The detailed criteria and specific rubrics for

each question type are presented as follows. These criteria served as guidelines for human expert annotators in the question selection process from the initial corpus of 932 questions. Our final result shows that inter-annotator agreement (Pearson Correlation) achieved r = 0.76, and only questions achieving average scores $\geq 4.5$ were retained.

---

**Technical Details Questions Evaluation Rubric**

**Key Evaluation Dimensions:**

- **Technical Specificity**: Whether the question targets specific technical concepts, implementations, or methodologies with clear scope and boundaries.

- **Precision Requirements**: Whether the question demands accurate, detailed explanations that can differentiate between correct and incorrect technical knowledge.

- **Factual Verifiability**: Whether the answer can be verified against authoritative sources (documentation, standards, publications) with objective criteria.

- **Application Value**: Whether the question effectively reflects the model's retrieval capability for fine-grained technical problems.

**Scoring Standards:**

- **5 points**: Question pinpoints specific technical details with clear scope; requires precise, accurate explanations that demonstrate deep technical understanding; answers are easily verifiable against authoritative sources and remain stable over time; excellently tests model's ability to retrieve and explain complex technical knowledge.

- **4 points**: Question targets specific technical aspects with good clarity; requires detailed explanations with minor ambiguity; answers are generally verifiable with stable technical foundations; effectively tests technical knowledge retrieval with room for minor improvements.

- **3 points**: Question addresses technical details but with moderate specificity; requires explanations that may allow some interpretation; answers are partially verifiable but may depend on context or evolving standards; adequately tests technical knowledge but could be more precise.

- **2 points**: Question lacks technical specificity or is too broad; explanations may be vague or surface-level; answers are difficult to verify objectively or depend heavily on time-sensitive information; limited effectiveness in testing precise technical knowledge.

- **1 point**: Question is technically vague or overly general; fails to demand specific technical knowledge; answers cannot be objectively verified or are highly dependent on rapidly changing information; ineffective for testing detailed technical understanding.

---

**Literature Review Questions Evaluation Rubric**

**Key Evaluation Dimensions:**

- **Research Direction Clarity and Survey Scope**: Whether the question provides clear research direction and can guide comprehensive literature comparison and synthesis across multiple perspectives and methodologies.

- **Literature Coverage Requirements**: Whether the question demands systematic exploration of key papers, major approaches, and important findings in the specified research area.

- **Verifiability and Stability**: Whether the research direction is well-established with literature that is easily retrievable and verifiable, maintaining relevance over time.

**Scoring Standards:**

---

- **5 points**: Question provides clear research direction with well-defined scope; guides comprehensive literature exploration across multiple dimensions; targets stable research area with abundant, easily accessible literature; excellent potential for meaningful survey output.

- **4 points**: Question offers generally clear direction with adequate scope definition; encourages broad literature coverage; targets established research area with good literature accessibility; solid foundation for comprehensive survey work.

- **3 points**: Question provides moderate direction clarity with acceptable scope; requires literature coverage but may lack depth requirements; targets reasonably stable research area with moderate literature accessibility; sufficient for basic survey objectives.

- **2 points**: Question direction is somewhat unclear or too narrow/broad; limited guidance for comprehensive literature exploration; targets area with limited or hard-to-access literature; marginal value for survey purposes.

- **1 point**: Question lacks clear research direction or proper scope definition; fails to guide systematic literature exploration; targets unstable area with poor literature accessibility; inadequate for quality survey work.

---

**Open Consulting Questions Evaluation Rubric**

**Key Evaluation Dimensions:**

- **Openness and Depth**: Whether the question encourages creative exploration of multiple perspectives and stimulates deep, multi-dimensional thinking beyond conventional approaches.

- **Forward-Looking Value**: Whether the question addresses emerging trends, future challenges, or strategic insights that provide meaningful guidance for research directions or industry development.

- **Conceptual Innovation Potential**: Whether the question can inspire novel viewpoints, creative problem-solving approaches, or innovative thinking that advance understanding in the field.

- **Balanced Grounding and Long-term Relevance**: Whether the question maintains reasonable connection to existing knowledge while avoiding over-dependence on transient trends, ensuring lasting value.

**Scoring Standards:**

- **5 points**: Question demonstrates exceptional openness that encourages creative exploration across multiple dimensions; addresses significant forward-looking challenges or strategic opportunities; inspires innovative thinking and novel approaches; maintains strong grounding in fundamental principles while offering lasting relevance beyond current trends.

- **4 points**: Question shows high openness with good potential for multi-perspective exploration; addresses meaningful future-oriented topics with strategic value; encourages innovative thinking with reasonable grounding; demonstrates good long-term relevance.

- **3 points**: Question provides moderate openness with some potential for creative exploration; addresses topics with acceptable forward-looking value; allows for some innovative thinking but may lack depth; maintains reasonable balance between innovation and grounding.

- **2 points**: Question offers limited openness with constrained exploration potential; addresses topics with minimal strategic or future value; provides little inspiration for innovative thinking; may be either too abstract or too tied to current trends.

> • **1 point**: Question lacks meaningful openness and fails to encourage creative exploration; addresses topics with little strategic value or future relevance; provides minimal potential for innovative thinking; either completely ungrounded or overly dependent on temporary trends.

### B.5 EXAMPLES OF QUESTION SELECTION

To illustrate the application of our evaluation framework, we present concrete examples below to demonstrate how questions are assessed across different categories and quality levels. These examples showcase the evaluation process in action, highlighting both high-quality questions suitable for benchmark inclusion and problematic questions that require significant improvement.

---

**Example 1**

**Question:** Compare the Transformer and Mamba model architectures, analyzing their performance and technical characteristics in different application scenarios. Based on the latest research, discuss the advantages and disadvantages of both models and their applicable scenarios.

**Question Type:** Literature Review

**Annotator:**

1. **Research Direction Clarity and Survey Scope**
   - **Score:** 4/5
   - **Explanation:** The question provides generally clear research direction with a well-defined comparative framework between two specific model architectures (Transformer and Mamba). However, the scope could be more precise regarding specific application domains, evaluation metrics, or time frame for "latest research."

2. **Literature Coverage Requirements**
   - **Score:** 5/5
   - **Explanation:** The question effectively demands systematic exploration of foundational papers on both architectures (Attention is All You Need for Transformers, Mamba papers), comparative studies and benchmarks and Recent advances and variants in both families.

3. **Verifiability and Stability**
   - **Score:** 3/5
   - **Explanation:** Transformer architecture is extremely well-established with abundant, easily accessible literature, and core architectural principles are stable and well-documented. However, Mamba is a relatively recent architecture (2023), so the literature base is still developing. Rapid evolution means some findings may become outdated quickly.

**Overall Assessment:**
**Overall Score:** 4/5
**Recommendation:** HIGH VALUE for inclusion
**Explanation:** This question demonstrates high benchmark value by combining real research relevance, comprehensive skill testing across multiple dimensions (synthesis, analysis, reasoning, application), balanced complexity, contemporary significance in architectural debates, and objective verifiability against established research standards.

---

**Example 2**

**Question:** What are the differences between GQA and MQA? And what are their respective applicable scenarios?

---

**Question Type:** Technical Details

**Annotator:**

1. **Technical Specificity**

   - **Score:** 2/5
   - **Explanation:** The question lacks sufficient technical specificity. Firstly, "GQA" is ambiguous - could refer to Grouped Query Attention or Graph Question Answering. Secondly, the scope is too broad, asking for general differences and applications, no specific implementation details, parameters, or technical boundaries are defined. Last but not least, context about which domain is missing (NLP, computer vision, etc.).

2. **Precision Requirements**

   - **Score:** 2/5
   - **Explanation:** The question has significant precision issues: The ambiguous "GQA" term makes precise explanation difficult, and respective application scenarios is too general. It doesn't demand specific technical details about attention mechanisms, computational complexity, or implementation specifics, and allows for surface-level explanations rather than deep technical understanding.

3. **Factual Verifiability**

   - **Score:** 3/5
   - **Explanation:** If GQA refers to Grouped Query Attention, it can be verified against recent research papers, while MQA is well-documented in transformer literature. However, the ambiguous terminology makes verification challenging.

4. **Application Value**

   - **Score:** 2/5
   - **Explanation:** Limited application value due to the ambiguity reduces its effectiveness in testing precise technical knowledge. This question doesn't effectively differentiate between surface knowledge and deep understanding, and it's too general to test fine-grained technical retrieval capabilities.

**Overall Assessment**
**Overall Score:** 2.25/5
**Recommendation:** LOW VALUE for inclusion
**Explanation:** This question should NOT be included due to fundamental ambiguity ("GQA" is unclear), poor discriminative power, verification challenges, and limited technical rigor that fails to effectively test precise technical knowledge.
To improve this question, it should clearly specify "Grouped Query Attention (GQA)" vs "Multi-Query Attention (MQA)", demand specific technical details (computational complexity, memory usage, implementation differences), and define clear evaluation metrics with concrete examples.

## C GUIDELINES FOR RUBRIC DESIGN

### C.1 KEY INSIGHT EXTRACTION

We firstly use `Claude-3.7-Sonnet` to analyze the contextual source material for each question, identifying key insights and generating comprehensive reference materials. For different question types, we design different focus areas when extracting key insights:

**Technical Details Questions - Key Insight Guidelines**

For technical details questions, the analysis focuses on extracting structured key insights with emphasis on:

1. Precise technical specifications and parameters
2. Detailed algorithmic descriptions and mathematical formulations
3. Implementation considerations and computational requirements
4. Performance metrics and efficiency analyses
5. Technical limitations and edge cases
6. Optimization techniques and fine-tuning procedures
7. Code examples or pseudocode where applicable
8. System architecture and component interactions
9. Technical dependencies and environmental requirements
10. Debugging approaches and common technical pitfalls

**Literature Review Questions - Key Insight Guidelines**

For literature review questions, the analysis focuses on extracting structured key insights with emphasis on:

1. Comprehensive overview of the technological landscape
2. Historical development and evolution of relevant technologies
3. Current state-of-the-art approaches and methodologies
4. Comparative analysis of different technical solutions
5. Key research papers, influential publications and bibliographic references
6. Emerging trends and future research directions
7. Major contributors and research groups in the field
8. Theoretical foundations and fundamental principles
9. Cross-disciplinary connections and applications
10. Benchmark datasets and evaluation frameworks commonly used in the field

**Open Consulting Questions - Key Insight Guidelines**

For open consulting questions, the analysis focuses on extracting structured key insights with emphasis on:

1. Provision of new insights beyond common knowledge or existing literature
2. In-depth analysis of the question from multiple perspectives
3. Critical thinking and identification of key challenges and core problems
4. Novel hypotheses, conceptual frameworks, or alternative viewpoints
5. Strategic discussions on potential research directions or practical solutions
6. Integration of cross-disciplinary knowledge to enrich the analysis
7. Reflection on the broader implications, including societal, ethical, and industrial impacts
8. Exploration of future trends and transformative opportunities
9. Expert judgment supported by logical reasoning and evidence
10. Creative and thought-provoking ideas that inspire further discussion

### C.1.1 KEY INSIGHT EXTRACTION PROMPT

We employ the following prompt to extract key insights from the contextual source material of questions, and generate auxiliary materials as reference.

**Key Insight Extraction Prompt Template**

<system_role>
You are an expert research analyst specializing in extracting high-value insights from academic and technical content.
</system_role>

<user_prompt>
Your task is to identify and structure key insights that demonstrate deep understanding and expert-level analysis. Given the following question and its contextual source material, extract key insights following the specific guidelines for {Question Type} questions.

**Question:** {Question}

**Source Material:** {Source Context}

**Guidelines:** {Guideline for Question Type}

Please identify and extract 8-15 key insights that represent the most valuable and insightful aspects of addressing this question. Each insight should be:

- Substantive and demonstrate deep understanding
- Directly relevant to answering the question
- Represent expert-level analysis or specialized knowledge
- Be specific enough to be evaluable

Format your response as a structured list of key insights with brief explanations.
</user_prompt>

**Reference Material Generation Prompt Template**

<system_role>
You are an expert research analyst specializing in extracting relevant information from the document and providing a comprehensive reference materials.
</system_role>

<user_prompt>
Your task is to:

- Carefully analyze the document script provided
- Extract all relevant information related to the specific question, containing all the key insights
- Organize the information in a coherent, well-structured response
- Provide accurate, helpful information based primarily on the document content

IMPORTANT: If the document and those key insights do not contain enough information to fully answer the question, you may supplement with your general knowledge, but you must clearly indicate which parts are from the document and which parts are your additional context or expertise. Always prioritize information from the document and provided key insight, and only add relevant knowledge when necessary to provide a more complete answer.

**Question:** {Question}

**Full Document:** {Document}

**Key Insights:** {Key Insights List}

Please provide a comprehensive answer to the question based primarily on the information in the document and key insights. If needed, you may supplement the answer with your own knowledge, but clearly distinguish between information from the document and your additional insights.

**</user_prompt>**

### C.1.2 HUMAN VERIFICATION AND RUBRIC DESIGN

Based on the extracted key insights and reference materials, we invite expert annotators to design corresponding rubrics for each question. We established the following annotation guidelines:

---

**Rubric Design Annotation Guidelines**

**Task Overview:**
You will be presented with a research question, a list of key insights, reference materials, and complete contextual information discussing this research question. Your task is to design rubric based on key insights for evaluating the quality of reports answering this research question, and assign weights to different criteria based on value judgment.

**Rubric Design Requirements:**

- **Clarity and Verifiability**: Each criterion must be clearly described and easy to verify objectively.

- **Independence**: Ensure each criterion is independently verifiable without overlap.

- **Conceptual Focus**: Focus on essential concepts rather than specific examples.

- **Evaluative Phrasing**: Phrase each criterion as an evaluative statement, not a descriptive one.

- **Objective Assessment**: Make each criterion specific enough to enable clear pass/-fail evaluation.

- **Action-Oriented Language**: Use verbs like "Explains," "Describes," "Discusses," "Outlines," "Provides," "Analyzes," "Compares," "Identifies" to indicate the expected level of detail and engagement with concepts.

- **Contextual Reference**: The reference materials serve only as auxiliary annotation text and may not accurately reflect the original discussion content. When confused, refer to the original context for specific information.

**Weight Assignment Guidelines:** Assign weights from 1-3 based on the importance of each criterion to answering the question comprehensively:

- Higher weights (3) should be assigned to criteria that are core to understanding the core question

- Medium weights (2) for supporting criteria that add depth and context

- Lower weights (1) for nice-to-have criteria that enhance but are not essential to the answer

- Ensure total weights reflect the relative importance hierarchy of different aspects

**Quality Control Measures:**

- Each criterion should be binary assessable (present/absent)

- Avoid subjective language that could lead to inconsistent evaluation

- Test each criterion against the reference materials to ensure it captures meaningful distinctions

- Ensure each criterion collectively cover the essential aspects of a comprehensive answer

---

## C.2 Rubric Example

To demonstrate the dynamic rubric generation process, we present examples showing how specific evaluation criterion are designed based on key insights extracted from responses.

**Question:** How should the development of generative AI evolve: focusing on dialogue-based systems (Chat) or autonomous action-taking systems (Agent)? What are the key differences, technological requirements, and future implications of each approach?

---

**Example 1**

**Key Insight:**
```
Agent evolution mirrors historical tech patterns like smart
speakers becoming home automation hubs and Robotic Process
Automation (RPA) transitioning to Agentic Process Automation (APA).
```
**Criterion:**
```
Provides relevant historical analogies to similar technological
evolutions (e.g., smart speakers, voice assistants).
```
(weight=1)

---

**Example 2**

**Key Insight:**
```
Chat systems focus on information processing and linguistic
exchange, operating as 'brain and mouth' entities confined to
conversational domains (e.g.  ChatGPT), while agent systems
represent a paradigm shift from predefined logic in traditional
software to autonomous, goal-oriented solutions generated
collaboratively between developers and AI.
```
**Criterion:**
```
Clearly distinguishes between process-oriented (Chat) and
goal-oriented (Agent) frameworks.
```
(weight=2)

---

The first criterion captures the importance of historical context and analogical reasoning, while the second focuses on the fundamental architectural distinctions between different AI paradigms.

# D    Experimental Setup

---

**Report Generation Prompt Template for Non-DARS systems**

Generate a comprehensive research report addressing the following research questions. Your report must include the clear structure, detailed explanations, and references to relevant academic sources. You should use IEEE style citations for all references, Use numbered citations in square brackets like [1], [2], [3].
Research Question: [QUESTION]

---

For most DARS systems except Perplexity Deep Research, interaction was only possible through web user interfaces (WebUI), which somewhat limited the scalability of our evaluation experiments. To ensure fairness across different DARS systems, we employed default settings when interacting with all systems. Specifically, OpenAI Deep Research, Claude Research and Doubao Deep Research typically asks follow-up questions after the initial user query, and our standard response was "By default." to maintain consistency across evaluations. Gemini-2.5-Pro Deep Research pre-generates research plans before conducting research, and we directly used these generated plans without any modifications to maintain system autonomy and avoid human intervention bias.

For non-DARS systems with web search capabilities (GPT-4o Search Preview and Perplexity: Sonar Reasoning Pro), we configured the Search Context Size to "High" setting to maximize their research capabilities and ensure fair comparison with dedicated research agents. This enhancement was

designed to compensate for the lack of specialized research workflows in these systems by providing them with expanded search context, thereby enabling more comprehensive information retrieval and analysis. Additionally, we designed a unified prompt for non-DARS systems to guide models in generating responses with proper report format.

# E   EXPERIMENTAL DETAILS

## E.1   DATA COLLECTION TIMEFRAMES

Due to the lack of transparency regarding model iterations and technical details in most commercial DARS systems, we explicitly documented the timeframes during which our data collection is taken in Table 5.

Table 5: Data Collection Timeframes for DARS

| DARS | Data Collection Timeframe |
|---|---|
| OpenAI Deep Research | March 24 – April 29 |
| Perplexity Deep Research | March 24 – April 15 |
| Grok Deep Search | March 25 – April 14 |
| Gemini Deep Research | April 15 – April 21 |
| Grok Deeper Search | April 18 – April 19 |
| Claude Search | July 21 – July 28 |
| Doubao Deeper Search | July 21 – July 28 |
| Mita Deeper Search | July 21 – July 30 |

## E.2   IMPLEMENTATION DETAILS

While our evaluation framework follows the methodology outlined in Section 4, several specific optimizations were adopted to enhance robustness and efficiency in practice.

**Rubric Assessment Implementation.**   For rubric assessment, we evaluate each criterion independently using the prompt template provided in Appendix E.3.

**Factual Assessment Implementation.**   For factual assessment, we implemented a section-based claim extraction strategy rather than processing entire reports simultaneously, considering computational efficiency and context length limitations. Specifically, reports are segmented into sections by paragraph boundaries, and claims are extracted from each section independently using the template in Appendix E.4. Finally, extracted claims from all sections are subsequently aggregated to form the complete claim set for each response.

Following extraction, claims sharing identical URL sources are grouped to optimize verification and reduce redundant web retrieval. Each URL group undergoes batch verification using the prompt in Appendix E.5. To handle potential failures and errors robustly, we extended the binary judgment framework into a three-way classification ("yes", "no", and "unknown"). This three-way classification allows the system to gracefully handle cases where claims are incompletely extracted or source URLs are inaccessible. Claims classified as "unknown" are excluded from final metrics to ensure evaluation reliability.

## E.3   RUBRIC COVERAGE EVALUATION PROMPT

> **Rubric Coverage Evaluation Prompt**
>
> ## Task
> Determine whether the AI response adequately covers the specific criterion provided. Answer with "yes" or "no" followed by a brief justification.

## Input Materials
<Question>: {question}
<Criterion>: {criterion}
<Weight>: {weight} (indicates the importance of this criterion)
<AI Response>: {ai_response}
## Evaluation Criteria

- Answer "yes" if the AI response clearly includes or adequately expresses the main content of the criterion

- Answer "yes" if the response conveys the same meaning as the criterion, even if using different terminology or phrasing

- Answer "no" if the AI response only partially addresses or completely fails to mention the content of the criterion

- Consider semantic equivalence, not just keyword matching

- Pay special attention to technical details, numerical values, and specific claims in the criterion

## Output Format
Your answer must begin with either "yes" or "no" followed by a brief justification.
## Example format
"yes: The response clearly addresses this criterion by explaining [specific detail]..."
"no: While the response mentions [related concept], it fails to address [specific aspect] of the criterion..."

### E.4 CLAIMS EXTRACTION PROMPT

**Claims Extraction Prompt**

## Task Description
Extract all factual claims from the provided academic paper. Each claim should be a factual statement that can be verified. Claims may or may not have supporting citations.
## Input
A Research Question and a complete academic paper containing factual claims, some of which may have citation markers and corresponding URLs (inline or in reference section).
## Output Requirements

- Extract each distinct factual claim throughout the entire paper

- For each claim, output a JSON object with:

  - The exact claim text as a string
  - The original text from the paper containing this claim (context)
  - The corresponding citation URL as source (if a citation marker directly follows the claim)

- If a claim has a citation marker directly following it, return the supporting URL

- If a claim have no citation marker directly following it, return an empty string

- Ensure all string values are properly escaped for valid JSON format (e.g. Replace internal quotation marks (") with escaped quotation marks (\"))

- Return a JSON array containing all claim objects

## Format Specification

```
[
  {
    "claim": "The exact statement representing a factual claim",
    "context": "The original sentence or passage from the paper
                containing this claim",
    "source": "https://example.com/source1"
  },
```

```
    {
      "claim": "Another factual statement without direct citation",
      "context": "The original sentence or passage from the paper
                  containing this claim",
      "source": ""
    }
]
```

## Guidelines for Claim Identification

1. A claim should be a complete, standalone factual statement

2. Maintain the original wording where possible, but remove unnecessary context

3. Extract all factual claims regardless of whether they have citation support

4. Only consider to map citation markers (numbers, author names, etc.) to their corresponding URLs in the references section when it directly follow the claim statement

5. Exclude opinions, speculations, or methodological descriptions

6. Extract the context passage containing each claim for verification purposes

7. If multiple claims are associated with the same citation, extract them as separate entries

## Citation URL Mapping

- If URLs appear directly after claims, use those URLs directly

- Citation markers (e.g. follows a number or [number]) must directly follow the claim to be considered as supporting that claim

- If claims use citation markers that reference a bibliography or reference section, locate the corresponding URLs in that section

- If a claim has no directly following citation marker, use an empty string for source

Please extract all claims from the response content and output in the specified JSON format:
Research Question: [QUESTION]
Response Content: [CONTENT]
References: [REFERENCES]

### E.5 CLAIM VERIFICATION PROMPT

**Claim Verification Prompt**

## Task Description
Your task is to verify whether claims are supported by the provided reference content.
## Input

- A reference content that contains supporting information

- A list of claim-context pairs that need to be verified against the reference

## Output
For each claim, respond with 'yes', 'no', or 'unknown' to indicate whether the claim is supported by the reference content. Output in the specified JSON format.
## Output Format Specification

```
[
  {
    "id": 1,
    "result": "yes"
  },
  {
    "id": 2,
    "result": "no"
  },
  {
```

```
    "id": 3,
    "result": "unknown"
  }
]
```
## Verification Guidelines
### Claim Support Determination
If the reference is valid, for each given claim:

- **'yes'**: If the facts or data in the claim can be found entirely or partially within the reference content

- **'no'**: If all facts and data in the statement cannot be found in the reference content

- **'unknown'**: If verification encounters difficulties (such as semantic incompleteness, ambiguity, or other issues that make verification impossible), or reference contains are not available ('page not found' message, connection errors, or other non-content responses)

Claims must be verifiable from the content provided, not based on general knowledge.
### Using Context Information
If you encounter difficulties when verifying claims (e.g., semantic incompleteness/ambiguity issues), refer to the corresponding additional context. If problems still exist after considering the paragraph context, output 'unknown'.
Please provide your verification results in the specified JSON format.
Source: [SOURCE]
Claim-Context Pair List: [CLAIM_LIST]

## E.6 ERROR ANALYSIS AND EXAMPLES

### E.6.1 TAXONOMY AND METHODOLOGY

To gain deeper insights into the failure modes of DARS, we perform a systematic error analysis on a representative subset of our dataset. We categorize common failure patterns along two dimensions: rubric assessment errors and factual assessment errors:

**Rubric Assessment Errors.** For responses that fail to satisfy any evaluation criteria, we identify two primary error categories:

- **Incompleteness**: The response completely fails to address or satisfy the criterion, demonstrating a fundamental gap in research coverage.
- **Partial Missing**: The response addresses only a subset of the criterion's requirements, indicating incomplete analysis or synthesis.

**Factual Assessment Errors.** For claims in the responses that fail verification, we also identify two primary error categories:

- **Invalid Attribution**: The cited URL is expired, malformed, fabricated, or otherwise inaccessible, preventing verification regardless of claim validity.
- **Unsupported Answer**: The URL contains relevant information but the model misrepresents, misinterprets, or hallucinates content not present in the source.

Human annotators examined responses from five representative DARS (OpenAI Deep Research, Gemini Deep Research, Claude Research, Doubao Deep Research, and Mita Deep Research), with 10 responses sampled from each system (50 responses in total). For rubric assessment, annotators directly categorized uncovered criteria according to the three-way taxonomy. For factual assessment, annotators first employed the same web retrieval API used in our main experiments to determine URL accessibility (Invalid Attribution), then compared model responses against retrieved content to identify hallucinations (Unsupported Answer).

| Error Type | Open Consulting | Literature Review | Technical Details |
|---|---|---|---|
| *Rubric Assessment* | | | |
| Incompleteness | 4.47 | 6.23 | 5.63 |
| Partial Missing | 1.80 | 2.64 | 2.93 |
| *Subtotal* | *6.27* | *8.87* | *8.56* |
| *Factual Assessment* | | | |
| Invalid Attribution | 2.48 | 2.35 | 1.42 |
| Unsupported Answer | 16.40 | 16.35 | 15.50 |
| *Subtotal* | *18.88* | *18.70* | *16.92* |

Table 6: Average number of errors per response across different question types. Error distributions reveal distinct failure patterns in rubric coverage versus factual grounding.

### E.6.2 RESULTS AND ANALYSIS.

Table 6 presents the average error count per response across question types. Several key patterns emerge from this analysis:

**Error Check for Rubric Assessment.** 1) Open consulting questions exhibit the lowest average error rate (6.27), corroborating our main findings in Section 5.2 regarding DARS's outstanding performance on exploratory research tasks. 2) Literature review questions show the highest rubric error rate (8.87), particularly in incompleteness (6.23), indicating that systematic coverage of research breadth remains challenging even for state-of-the-art DARS. 3) Incompleteness errors dominate partial missing errors across all question types (2.5:1 of rubric errors on average), suggesting that DARS demonstrate strong integration and reasoning capabilities for information they successfully retrieve, but struggle to answer when relevant information is not retrieved during the search phase. This indicates that retrieval capability might represent a primary bottleneck in rubric assessment.

**Error Check for Factual Assessment.** 1) DARS demonstrate higher hallucination rates in open consulting (16.40) and literature review questions (16.35) compared to technical details questions (15.50). This pattern suggests that when synthesizing broad conceptual insights or surveying literature landscapes, DARS are more prone to misrepresenting source content than when retrieving specific technical information. 2) Failure modes in the factual assessment varies significantly across different systems. For instance, Mita Deep Research exhibits a strong bias due to inaccessible URLs (Invalid Attribution : Unsupported Answer = 2.32 : 15.56), suggesting retrieval quality issues in its web search engine. Conversely, Doubao Deep Research demonstrates a different failure profile (Invalid Attribution : Unsupported Answer = 1.98 : 16.36), with hallucination problems primarily stemming from the model's content generation rather than source accessibility.

### E.6.3 EXAMPLE CASES OF ERROR TYPES

To illustrate the error patterns identified in our analysis, we present representative examples from our annotated dataset. These cases demonstrate the specific manifestations of each error category and provide actionable insights into future DARS system improvements.

---

**Incompleteness Error Example: Claude Research**

**Question:** What new types of 'creative infrastructure' does the web need to support AI-generated 3D/immersive experiences while maintaining open standards?

**Criterion:** Discuss how the transition from Developer Experience (DX) to Agent Experience (AX) represents a fundamental shift in web development philosophy for supporting autonomous content generation.

**Error Analysis:** The response never engaged with the specific conceptual shift from Developer Experience (DX) to Agent Experience (AX). Instead, it focused solely on technical and infrastructural challenges of AI-generated 3D web content, ignoring the rubric's requirement

---

to discuss how AX fundamentally redefines web development philosophy for autonomous content generation.

---

**Partial Missing Error Example: Gemini Deep Research**

**Question:** In multimodal pretraining, the current mainstream paradigms are based on image tokens and stable diffusion. Analyzing the latest advancements (by April 2025) in these two technical approaches, with reference to the most recent papers, which one appears to be more promising and why?

**Criterion:** Explains how compressed latent space operation balances computational efficiency with detail preservation in recent stable diffusion approaches.

**Error Analysis:** The model response gestured at the idea of operating in a compressed latent space for efficiency but stopped short of explicitly addressing the core trade-off that the rubric required: how recent stable diffusion methods balance computational gains with the preservation of fine-grained detail. It mentioned latent diffusion and compression, but failed to connect this to concrete aspects of detail preservation.

---

**Invalid Attribution Error Example: Doubao Deep Research**

**Question:** "Why can models trained on synthetic data outperform the models that provide the synthetic data? Please find the latest research papers..."

**Claim:** "Structured Domain Randomization outperformed real data on the KITTI object detection task in Computer Vision."

**Cited Source:** $https://m.zhangqiaokeyan.com/academic-conference-foreign-international-conference-robotics-and_thesis/020512349961.html$

**Source Status:** Invalid URL (403 Forbidden)

**Error Analysis:** The cited URL returns a 403 Forbidden error and does not provide access to the source content. Therefore, it is impossible to verify or refute the claim regarding Structured Domain Randomization's performance on the KITTI object detection task.

---

**Unsupported Answer Error Example: Mita Deep Research**

**Question:** What is the fundamental reason behind the low cost of DeepSeek V3? Is it due to leveraging data distillation from other 'teacher models' (such as OpenAI, Gemini, etc.), or adjustments in training and inference precision algorithms?

**Claim:** "DeepSeek-V3's inference pricing is at 1% of GPT-4o's rates."

**Cited Source:** $https://arxiv.org/pdf/2412.19437$

**Source Title:** DeepSeek-V3 Technical Report

**Source Status:** Accessible

**Error Analysis:** The claim states that DeepSeek-V3's inference pricing is at 1% of GPT-4o's rates. The retrieved source provides extensive details on DeepSeek-V3's architecture, training costs, and performance, but does not mention or quantify inference pricing relative to GPT-4o. There is no explicit comparison or percentage given for inference costs between DeepSeek-V3 and GPT-4o in the document.

### E.7    PAIRWISE COMPARISON ANALYSIS

For experimental rigor, we conducted pairwise comparison to complement our rubric-based assessment. We designed the evaluation prompt following the pairwise comparison methodology from the LLM-as-a-Judge variations proposed in (Zheng et al., 2023). Our pairwise comparison approach involves systematically comparing responses from different DARS models to the same questions, using a judge model to determine which response is better.

### E.7.1 METHODOLOGY

We conducted pairwise comparisons across all evaluated DARS systems. For each pair of models (Model A and Model B), we prepared their responses to all 65 questions for comparison. The judge model then evaluates each pair of responses side-by-side to determine the winner.

To mitigate potential positional bias that might arise from the presentation order of responses, we implemented a dual evaluation strategy: for each question, we evaluated both "Model A vs Model B" and "Model B vs Model A" comparisons. The final verdict for each question is determined by aggregating results from both evaluation directions, ensuring that presentation order does not influence the judgment outcome.

### E.7.2 PROMPT DESIGN

The evaluation prompt is designed to assess responses based on deep research characteristics, prioritizing accuracy, depth, comprehensiveness, and technical rigor. The specific prompt template is provided below:

---

**Prompt Template for pairwise comparisons**

Please act as an impartial judge and evaluate the quality of the responses provided by two AI research assistants to a deep research question displayed below. This is a Deep AI Research System (DARS) evaluation where you should assess which assistant provides a better research-quality response.
Your evaluation should prioritize the following deep research characteristics:

- **Accuracy and Factual Correctness**: The response should be factually accurate, well-cited (if applicable), and free from hallucinations or misinformation

- **Depth and Insight**: The response should demonstrate deep understanding, critical thinking, and meaningful insights rather than surface-level information

- **Comprehensiveness**: The response should cover all necessary aspects, address complexities, and consider multiple perspectives or approaches

- **Cross-domain Synthesis**: The response should effectively synthesize information across different domains, methodologies, or research areas when relevant

- **Frontier Research Awareness**: The response should demonstrate awareness of cutting-edge research, emerging directions, and open problems in the field

- **Technical Rigor**: For technical questions, the response should show appropriate technical depth, precision, and understanding of underlying principles

- **Practical Relevance**: The response should connect theoretical understanding with practical implications and real-world applications when appropriate

Begin your evaluation by comparing the two responses across these dimensions and provide a short explanation. Avoid any position biases and ensure that the order in which the responses were presented does not influence your decision. **Do not allow the length of the responses to influence your evaluation - prioritize quality and depth over quantity.** Do not favor certain names of the assistants. Be as objective as possible.
After providing your explanation, output your final verdict by strictly following this format: "[[A]]" if assistant A is better, "[[B]]" if assistant B is better, and "[[C]]" for a tie.
**[Research Question]** {question}
**[The Start of Assistant A's Response]** {response_a} **[The End of Assistant A's Response]**
**[The Start of Assistant B's Response]** {response_b} **[The End of Assistant B's Response]**

---

### E.7.3 EXPERIMENTAL RESULTS

We evaluated all pairwise combinations across the 10 evaluated systems (8 DARS and 2 LLM systems with web search tools), resulting in 45 model pairs. Each pair comparison was conducted on all 65 questions. Table 7 presents detailed statistics for each model across all pairwise comparisons.

| Model | Pairwise Comparison | | | Win Rate | Avg. Length (chars) |
|-------|------|------|------|----------|---------------------|
| | **Win** | **Tie** | **Lose** | | |
| Gemini Deep Research | 527 | 43 | 15 | **90.09%** | 78, 857 |
| Doubao Deep Research | 431 | 85 | 69 | 73.68% | 58, 453 |
| OpenAI Deep Research | 401 | 84 | 100 | 68.55% | 49, 335 |
| Mita Deep Research | 317 | 125 | 143 | 54.19% | 53, 189 |
| Claude Research | 245 | 109 | 231 | 41.88% | 15, 183 |
| Perplexity Deep Research | 195 | 103 | 287 | 33.33% | 13, 113 |
| Grok3 DeeperSearch | 156 | 107 | 322 | 26.67% | 14, 132 |
| Grok3 DeepSearch | 141 | 121 | 323 | 24.10% | 14, 380 |
| Perplexity: Sonar Reasoning Pro | 40 | 52 | 493 | 6.84% | 4, 850 |
| GPT-4o Search Preview | 34 | 47 | 504 | 5.81% | 6, 718 |

Table 7: Pairwise comparison results showing detailed statistics for each system across all pairwise comparisons. Top performance is highlighted in bold.

### E.7.4  COMPARISON WITH RUBRIC ASSESSMENT RESULTS

Comparing the pairwise comparison results with our rubric assessment results from Section 5.2 (Table 3), we observe several notable discrepancies:

1. **OpenAI and Claude Performance Decline:** In pairwise comparison, OpenAI Deep Research achieves a win rate of 68.55%, ranking third, compared to its top-ranked coverage score of 0.7032 in rubric assessment. Similarly, Claude Research shows a more significant decline, with a win rate of 41.88% in pairwise comparison, ranking fifth, versus a coverage score of 0.6113 (ranking third) in rubric assessment.

2. **Doubao Performance Increase:** Doubao Deep Research demonstrates a substantial improvement in pairwise comparison, achieving a win rate of 73.68% and ranking second, compared to its coverage score of 0.5754 (ranking fifth) in rubric assessment.

3. **Gemini Maintains Top Position:** Gemini Deep Research consistently performs best in both evaluation methods, achieving the highest win rate (90.09%) in pairwise comparison and the second-highest coverage score (0.6929) in rubric assessment.

### E.7.5  LENGTH BIAS ANALYSIS

To investigate the cause of these discrepancies, we analyzed the correlation between response length and pairwise comparison win rates. Table 7 and statistical analysis reveal a **strong positive correlation** between average response length and win rate. Statistical analysis demonstrates that response length and win rate exhibit an extremely strong positive correlation:

- **Pearson correlation coefficient:** $r = 0.9487$ ($p < 0.001$, highly significant)
- **Spearman rank correlation coefficient:** $\rho = 0.9273$ ($p < 0.001$, highly significant)
- **Coefficient of determination:** $R^2 = 0.90$, indicating that response length explains **90% of the variance** in win rates

Despite explicit instructions in the prompt to "**Do not allow the length of the responses to influence your evaluation - prioritize quality and depth over quantity**", the judge model exhibits a strong bias favoring longer responses. This bias likely emerges indirectly through the evaluation criteria: when assessing dimensions such as "Comprehensiveness" and "Depth and Insight", longer responses may appear more comprehensive or detailed simply due to their greater volume, even if the additional content lacks substantive value.

### E.7.6  DISCUSSION AND IMPLICATIONS

The observed length bias in pairwise comparison raises significant concerns about its reliability for assessing DARS quality. Our statistical analysis reveals an extremely high correlation between response length and win rates ($r = 0.9487$, $R^2 = 0.90$), indicating that pairwise comparison results are primarily driven by response length rather than actual research quality.

In contrast, our rubric-based assessment methodology effectively mitigates this length bias problem. The rubric evaluation framework assesses responses based on whether they address specific, expert-designed criteria. Each criterion is evaluated independently using binary judgment (satisfied/not satisfied), and only content that directly addresses the criterion contributes to the score. Under this framework, **longer responses do not automatically receive higher scores**; responses must specifically address the expert-designed criteria to earn points, regardless of their overall length. Even if a response is extremely long, it will not score well unless it effectively addresses the rubric criteria, as irrelevant or redundant content does not contribute to the coverage score.

This fundamental difference explains the discrepancies between pairwise comparison and rubric assessment results. Models like Doubao, which generate longer responses, benefit significantly in pairwise comparison due to the length bias, while models like OpenAI and Claude, which provide more concise but high-quality responses, are penalized. The rubric-based approach, by focusing on criterion-specific content rather than overall comprehensiveness, provides a more reliable and objective assessment of actual research insight quality.

# F   CASE STUDY

We conduct a case study to demonstrate how different operational modes produce distinct value in research outputs. We identify three primary response modes: 1) **Information Retrieval Mode**: directly utilizing retrieved information to answer rubric criteria, reflecting high-quality source filtering and information retrieval capabilities; 2) **Internal Knowledge Mode**: leveraging the model's internal knowledge base to answer rubric criteria, requiring strong model capabilities; and 3) **Reasoning-based Synthesis Mode**: performing inference based on retrieved information to answer rubric criteria, demonstrating DARS's information integration and analytical capabilities.

We select the third criterion of Question 51 as example to analyze different DARS response modes:

---

**Example of Question and Evaluation Criterion**

**Question:** "What is the conceptual difference between Mixture of Experts (MoE) in Large Language Models versus traditional recommendation systems, and why do LLMs process tokens rather than entire sentences through individual experts?"

**Criterion:** "Explains at least one computational efficiency benefit of token-level processing versus sentence-level processing in LLMs."

---

The following presents response fragments from different DARS that correctly address this criterion, categorized by their operational modes:

---

**Information Retrieval Mode: Perplexity Deep Research**

**Response:** "**Training Speed**: MoE LLMs with token-level routing have faster training and better or comparable performance than dense LLMs on many benchmarks. Kirakosyan (2025)"

**Analysis:** The response directly quotes retrieved information without substantial reasoning. It precisely locates and extracts key information from external sources, specifically citing the original text from the Neptune.ai blog on Mixture of Experts LLMs.

---

**Internal Knowledge Mode: GPT-4o Search Preview**

**Response:** "**Computational Efficiency**: By focusing on tokens, LLMs can apply parallel processing techniques, enhancing computational efficiency and scalability."

**Analysis:** The explanation leverages fundamental knowledge about parallel processing in Transformer architectures. This content is not cited and 'parallel processing' is not mentioned in the retrieved sources, indicating the model draws from its internal knowledge base rather than real-time web retrieval.

---

---

**Single-source Reasoning-based Synthesis Mode: Grok3 DeeperSearch**

**Response:** "**Computational Efficiency and Load Balancing:** Token-level assignment facilitates better load balancing across experts. Since experts can process multiple tokens from different sentences in parallel, it ensures efficient utilization of computational resources. If entire sentences were assigned to experts, longer or more complex sentences could overload specific experts, leading to imbalance. For instance, in Switch Transformers, the top-2 routing ensures that experts are utilized evenly, enhancing efficiency. (Fedus et al. (2022))"

**Analysis:** The model constructs its explanation using retrieved information as supporting evidence. It introduces "load balancing" as a key advantage and references the Switch Transformers paper to support this inference, demonstrating reasoning over retrieved content.

---

**Multi-source Reasoning-based Synthesis Mode: Gemini Deep Research**

**Response:** "**Computational Efficiency via Conditional Computation:** The core promise of MoE is reducing the computational cost (FLOPs) compared to a dense model with the same total parameter count. (Cai et al. (2025)) Token-level routing enables this by activating only a small subset of experts (e.g., 1 or 2 out of 8 or more) for each token. (Wolfe (2024)) This sparse activation drastically cuts down the computation required per token during both training and inference. (Sanseviero et al. (2023))"

**Analysis:** The model synthesizes information from three distinct sources: drawing the core MoE concept from an arXiv survey, explaining the mechanism via Stack Overflow, and reinforcing outcomes through Hugging Face documentation. This demonstrates comprehensive cross-source knowledge integration.

---

# G   HUMAN-DESIGNED VS LLM-GENERATED RUBRICS: A COMPARATIVE ANALYSIS

As discussed in Section 4, traditional LLM-as-a-Judge evaluation with single overall ratings provides insufficient granularity for assessing complex research responses. Frontier research evaluation requires nuanced assessment across multiple dimensions that single scores cannot capture. To address this limitation, ResearcherBench employs a human-in-the-loop three-stage method, engaging domain experts to construct fine-grained rubrics with specific criteria and assigned weights.

However, this expert-driven approach incurs higher annotation costs compared to fully automated methods. A critical question therefore arises: **can the current LLMs reliably automate this rubric construction process?** While we posit that automated systems struggle to identify valid evaluation dimensions for frontier AI problems, it is essential to provide empirical justification for this claim. In this section, we aim to demonstrate that the expert-designed approach yields tangible performance improvements — specifically regarding alignment with human evaluation — thereby justifying the necessity of human expertise over lower-cost automated baselines.

## G.1   EXPERIMENTAL DESIGN

We designed a comparative experiment that evaluates our expert-designed rubrics against two distinct types of LLM-generated baselines. These baselines represent the most common strategies for automated evaluation:

1. **Rubric$_{\text{dim}}$**: Rubrics generated based on four predefined dimensions (Comprehensiveness, Insight, Instruction-Following, and Readability), following the approach used in DeepResearch Bench (Du et al., 2025). This baseline represents a common approach where LLMs generate evaluation criteria based on general quality dimensions.

2. **Rubric$_{\text{ref}}$**: Rubrics constructed following the three-stage method detailed in Section 4, but substituting human experts with Claude-3.7-Sonnet in the Criteria Design (Stage 2) and Quality Control (Stage 3) phases. Specifically, we provided the LLM with identical reference materials, and adapted the human annotation guidelines into prompts to generate

evaluation criteria and weights (Stage 2), followed by a self-review process to refine the rubric quality (Stage 3). This baseline tests whether LLMs can effectively replicate the human expert's rubric design process when given the same reference materials.

For both baseline methods, we used the same judge model (o3-mini) and evaluation pipeline as our rubric assessment experiment to ensure fair comparison. We evaluated all DARS responses using these alternative rubrics and compared the results against our expert-designed rubric evaluation.

## G.2 COVERAGE SCORE RESULTS AND ANALYSIS

| Model | Rubric$_{dim}$ | Rubric$_{ref}$ | Expert-Designed |
|---|---|---|---|
| *Deep AI Research Systems* | | | |
| OpenAI Deep Research | 0.9693, [-0.010, +0.010] | 0.6527, [-0.052, +0.050] | 0.7032, [-0.050, +0.049] |
| Gemini Deep Research | 0.9792, [-0.017, +0.012] | 0.6706, [-0.055, +0.054] | 0.6929, [-0.053, +0.052] |
| Claude Research | 0.9635, [-0.011, +0.010] | 0.5616, [-0.052, +0.052] | 0.6113, [-0.053, +0.053] |
| Mita Deep Research | 0.9706, [-0.008, +0.008] | 0.5628, [-0.056, +0.055] | 0.5835, [-0.057, +0.059] |
| Doubao Deep Research | 0.9564, [-0.014, +0.012] | 0.5130, [-0.055, +0.054] | 0.5754, [-0.057, +0.055] |
| Perplexity Deep Research | 0.9552, [-0.012, +0.011] | 0.3921, [-0.050, +0.050] | 0.4800, [-0.056, +0.055] |
| Grok3 DeepSearch | 0.9390, [-0.022, +0.019] | 0.4074, [-0.051, +0.052] | 0.4414, [-0.054, +0.057] |
| Grok3 DeeperSearch | 0.9526, [-0.014, +0.013] | 0.4021, [-0.052, +0.052] | 0.4398, [-0.056, +0.056] |
| *LLM with Search Tools* | | | |
| Perplexity: Sonar Reasoning Pro | 0.9358, [-0.019, +0.017] | 0.4111, [-0.052, +0.053] | 0.4663, [-0.056, +0.056] |
| GPT-4o Search Preview | 0.8734, [-0.036, +0.028] | 0.2813, [-0.044, +0.045] | 0.3576, [-0.051, +0.052] |

Table 8: Comparison of coverage scores across Rubric$_{dim}$, Rubric$_{ref}$, and Expert-Designed Rubrics. We also report 95% bootstrap confidence intervals using 10,000 bootstrap samples.

Table 8 presents a comparative analysis of coverage scores across three distinct evaluation protocols: Expert-Designed Rubrics, Rubric$_{dim}$, and Rubric$_{ref}$. The quantitative results highlight substantial disparities in the discriminative power of each method:

**Rubric$_{dim}$ exhibits severe ceiling effects.** All systems achieve extremely high scores (ranging from 87% to 98%), with a minimal performance gap of only 10.58% between the highest and lowest performing models. This lack of variance indicates that criteria generated purely from generic metrics are overly lenient. They fail to capture nuanced quality differences, effectively categorizing almost all responses as "excellent" and rendering the evaluation ineffective for benchmarking purposes.

**Rubric$_{ref}$ suffers from misalignment and unreasonable design.** While this baseline demonstrates reasonable discriminative ability with a performance gap of 38.93%, the absolute scores are relatively low (ranging from 28% to 67%), compared to results of the expert-designed rubrics. This pattern suggests that without expert guidance, the LLM-generated criteria may be over rigid or misaligned with the core intent of the research questions. Consequently, the resulting evaluation fails to accurately measure the relevance and quality of the responses.

## G.3 HUMAN EVALUATION ALIGNMENT RESULTS AND ANALYSIS

To validate the reliability of different rubric generation methods, we assessed their alignment with human expert judgment using the methodology established in Section 4.3. Domain experts independently annotated a stratified subset of 20 responses using the criteria generated by each method. We then compared these human labels against the automated verdicts provided by the judge model (o3-mini). Table 9 details these alignment metrics — Accuracy, Precision, Recall, and F1-score — for each rubric generation method. The case study is presented in AppendixG.5.

The analysis reveals distinct characteristics for each rubric generation method:

**Rubric$_{dim}$ exhibits an artificially inflated F1-score of 0.95.** This high score reflects systematic bias rather than genuine alignment. The judge model marks nearly all criteria as satisfied, indicating that the generic criteria are excessively lenient and fail to make meaningful quality distinctions.

| Rubric Type | Acc. | Prec. | Rec. | F1 |
|---|---|---|---|---|
| Rubric$_{dim}$ | 0.91 | 0.91 | 0.99 | 0.95 |
| Rubric$_{ref}$ | 0.66 | 0.83 | 0.60 | 0.70 |
| Expert-Designed | 0.76 | 0.85 | 0.76 | 0.80 |

Table 9: Human evaluation alignment metrics for different rubric types.

**Conversely, Rubric$_{ref}$ achieves an F1-score of 0.70 with a notably low Recall of 0.60.** This indicates a high rate of false negatives, where the model penalizes responses that human experts consider valid. This pattern suggests that LLM-generated criteria are overly rigid: they often demand specific details present in the reference materials, failing to recognize valid alternative approaches or closely related technologies that effectively address the problem.

**In contrast, Expert-Designed rubrics achieve a balanced F1-score of 0.80.** This substantial agreement with human experts (Landis & Koch, 1977) demonstrates that expert-designed criteria effectively capture the nuanced judgment required for frontier research evaluation, striking the necessary balance between rigor and flexibility.

### G.4 WEIGHT DISTRIBUTION ANALYSIS

We analyzed the weight distribution patterns across different rubric types to understand how importance assignment differs. Table 10 presents the weight distribution for expert-designed rubrics versus LLM-generated rubrics.

| Weight | Expert-Designed | Rubric$_{dim}$ | Interpretation |
|---|---|---|---|
| 1 (Nice-to-have) | 35.45% | 73.66% | Criteria that's beneficial but not essential |
| 2 (Supporting) | 50.91% | 17.41% | Criteria that add depth and context |
| 3 (Essential) | 13.64% | 8.93% | Criteria core to understanding the question |

Table 10: Weight distribution comparison.

To further analyze the quality of LLM-generated rubrics, we manually assigned weights to Rubric$_{dim}$ following our Weight Assignment Guidelines in Appendix C. According to these guidelines, high weights (3) should be assigned to criteria that are core to understanding the core question, medium weights (2) for supporting criteria that add depth and context, and low weights (1) for nice-to-have criteria that's beneficial but not essential.

After applying these weight assignment standards to Rubric$_{dim}$, we discovered a problematic distribution pattern: Criteria with weight 1 and 2 dominate (73.66% and 17.41%, respectively), while criteria with weight 3 are severely underrepresented (only 8.93%). This distribution reveals some fundamental limitations in LLM-generated rubric quality:

**Insufficient weight 3 (essential criteria):** The scarcity of criteria with weight 3 (8.93%) means LLM-generated rubrics fail to emphasize the most critical aspects of frontier AI research questions. The deficiency leads to evaluations that cannot effectively distinguish between "adequate" and "excellent" responses, as the rubric fails to prioritize the most important evaluation dimensions. Without sufficient criteria with weight 3, the evaluation cannot focus on what truly matters for answering the research question comprehensively.

**Insufficient weight 2 (supporting criteria):** The scarcity of criteria with weight 2 (17.41%) indicates a lack of depth assessment. The deficiency in criteria with weight 2 means LLM-generated rubrics largely miss dimensions that assess the depth and comprehensiveness of responses beyond basic requirements. This results in evaluations that cannot distinguish between surface-level and in-depth analyses, as the rubric lacks criteria that assess the supporting details and contextual richness that characterize high-quality research responses.

**Over-abundant weight 1 (nice-to-have criteria):** The dominance of criteria with weight 1 (73.66%) indicates that LLMs tend to generate "safe" criteria that are easily satisfied but contribute little to distinguishing answer quality. The over-abundance means that evaluation is dominated by

peripheral aspects rather than core requirements. This explains the observed ceiling effects: most responses achieve artificially high scores by satisfying numerous low-importance criteria, masking meaningful differences in actual research quality.

A possible explanation for this phenomenon is that **LLMs lack the domain expertise necessary to identify which aspects of a frontier research question are truly core versus peripheral**. Without deep understanding of AI research priorities and evaluation standards, LLMs default to generating easily satisfiable criteria that avoid making strong claims. This results in a weight distribution that prioritizes peripheral enhancements over core understanding, fundamentally undermining the rubric's ability to assess research quality effectively.

### G.5 CASE STUDIES

To illustrate the qualitative differences between rubric types, we present case studies examining specific evaluation criteria.

### G.5.1 CASE STUDY: RUBRIC_DIM'S LACK OF DISCRIMINATIVE POWER

---

**Case Study 1: Superficial Evaluation Criteria (Rubric_dim)**

**Criterion:** "Formatting, Layout, and Highlighting for Readability"
**Analysis:** This criterion focuses entirely on surface-level presentation rather than the substance of the research content. It fails to impose specific requirements related to the research question, allowing models to score highly merely by producing well-formatted text, regardless of whether the actual technical content is accurate or deep.

---

**Case Study 2: Vague and Subjective Criteria (Rubric_dim)**

**Criterion:** "Originality and Synthesis in the Overall Approach"
**Analysis:** This criterion relies on abstract qualitative terms without defining specific benchmarks or expected outcomes. It is difficult to objectively quantify "originality" for a research summary task, leading to inconsistent evaluation. The lack of concrete indicators means the judge model often defaults to a positive score as long as the response reads smoothly, failing to distinguish true insight from generic summarization.

---

### G.5.2 CASE STUDY: RUBRIC_REF'S OVER-STRICTNESS

---

**Case Study 3: Overly Rigid Alignment (Rubric_ref)**

**Criterion:** "Explains how DeepSeek's combined advances (MoE, MLA, FP8, DualPipe, RL) shift industry trends toward algorithmic/hardware co-design and cost-aware scaling rather than brute-force parameter growth."
**Analysis:** This criterion was constructed as a strict checklist of specific terms derived from the reference materials, explicitly requiring the mention of "DualPipe." Therefore, the evaluation penalized a substantially correct response — which successfully discussed MoE, MLA, FP8, RL, and the core concept of cost efficiency — simply for missing a single keyword.

---

### G.6 DISCUSSION AND CONCLUSIONS

Our comparative analysis demonstrates that expert-designed rubrics are essential for evaluating frontier AI research questions. The key findings are:

1. **LLM-generated rubrics based on generic metrics fail to discriminate.** Rubric_dim exhibits severe ceiling effects, with all systems scoring between 87% and 98%, rendering the evaluation ineffective for benchmarking purposes.

2. **LLM-generated rubrics based on reference materials are overly strict and misaligned.** Rubric$_{\text{ref}}$ demonstrates better discriminative power but suffers from a low recall of 0.60, indicating that the criteria are overly rigid and fail to recognize valid and equivalent content.

3. **Expert-designed rubrics achieve optimal balance.** Expert-designed rubrics maintain strong discriminative power while achieving a balanced precision-recall profile (F1=0.80), demonstrating substantial alignment with human expert judgment.

4. **Weight distribution reveals fundamental limitations.** LLM-generated rubrics are dominated by peripheral criteria (73.66% criteria with weight 1) and lack sufficient core criteria (8.93% criteria with weight 3). This imbalance indicates that LLMs cannot effectively identify the essential elements of frontier AI research evaluation.

These findings support our claim that expert-designed rubrics are necessary for reliable evaluation of frontier AI research questions. The domain expertise required to identify core evaluation dimensions, balance strictness with flexibility, and assign appropriate importance weights cannot be reliably replicated by current LLM-based approaches.

## H   THE USE OF LARGE LANGUAGE MODELS

In the process of drafting this paper, we employed large language models (LLMs) as an auxiliary tool to enhance the quality and clarity of our written English. The primary application was to identify and correct grammatical inaccuracies, refine sentence structures, and polish academic expressions, thereby improving the overall readability and professionalism of the manuscript.

Specifically, selected paragraphs or sentences from our initial drafts were input into an LLM (e.g., DeepSeek-v3.1 or a comparable model) with explicit instructions focused solely on language checking and polishing. The prompts were designed to request grammatical corrections, suggestions for more concise or academically appropriate phrasing, and improvements in logical flow, without altering the core technical content or scientific meaning.

It is crucial to emphasize that the role of the LLM was strictly limited to writing assistant. All substantive intellectual contributions, including the core ideas, theoretical framework, experimental design, data analysis, and result interpretation, remain entirely our own. The final decision to adopt any suggestion provided by the LLM was always subject to our careful review and judgment. We ensured that every change aligned with our intended meaning and adhered to the standards of academic integrity.

This use of LLMs significantly streamlined the writing and revision process, allowing us to focus more effectively on the scientific rigor and conceptual depth of our work.

