# OpenReview forum: "ResearcherBench: Evaluating Deep AI Research Systems on the Frontiers of AI Research"
_ICLR.cc/2026/Conference — Submitted to ICLR 2026_

### Official Review · Reviewer_oGgf · 2025-10-28

**Soundness:** 2
**Presentation:** 2
**Contribution:** 2
**Rating:** 4
**Confidence:** 3

**Summary:**

This paper introduces ResearcherBench, a new benchmark for evaluating AI research systems on the task of understanding and extracting meaningful insights in research papers. ResearcherBench consists of 65 research questions across 35 AI subjects. The authors also propose a new evaluation pipeline that makes use of the proposed dataset involving expert-annotated evaluation criteria tailored to each research question.

**Strengths:**

* This paper introduces a dataset that includes 65 high-quality research questions. The authors asked expert evaluators to assess 932 research questions and selected 65 of them to ensure that the dataset contains only high-quality research questions.

* The proposed dataset includes expert-designed criteria for each research question. This is a clear advantage over existing datasets that expect a static rubric for evaluation.

**Weaknesses:**

My main concern lies in the novelty of the task and evaluation framework when compared to existing studies on related tasks such as report generation [1].

* **Novelty of the task.** This paper claims that "existing benchmarks primarily evaluate these systems on web retrieval and report generation abilities, overlooking their potential for discovering insights in AI research." However, it is unclear how the studied task substantially differs from the well-studied tasks like survey or report generation, which also often require extracting insights from papers.

* **Novelty of the evaluation framework.** The proposed evaluation pipeline using expert-curated criteria is reasonable, and I appreciate the annotation efforts. However, it appears to be similar to evaluation methods based on manually created references [1], and its advantages are unclear.

* **Rationale behind the dataset curation.** The authors claim to target "frontier AI research questions," but they do not systematically explain how the research questions in the introduced dataset are similar to or different from those in previous studies.

### Minor Limitations

* **Dataset size.** The introduced dataset includes only 65 research questions. While I understand the high cost of dataset creation due to expert annotation, the paper should still justify that a meaningful evaluation can be achieved with only 65 questions.

* **Missing human performance.** Including human performance in Table 2 would be helpful to assess how capable existing systems are for each task.

### Minor Comments

* I recommend using \citep when suitable.

[1] Mingxuan Du, Benfeng Xu, Chiwei Zhu, Xiaorui Wang, and Zhendong Mao. DeepResearch Bench: A Comprehensive Benchmark for Deep Research Agents. https://arxiv.org/abs/2506.11763, 2025.

**Questions:**

I expect responses to the points I listed in the first part of the Weaknesses section.

---

> ### Author Response · Authors · 2025-11-20
> **Official Comment by Authors**
>
> Thanks for reviewer's comments. Here are our responses to the comments.
>
> **Comment 1 (Novelty of the task):** This paper claims that "existing benchmarks primarily evaluate these systems on web retrieval and report generation abilities, overlooking their potential for discovering insights in AI research." However, it is unclear how the studied task substantially differs from the well-studied tasks like survey or report generation, which also often require extracting insights from papers.
>
> **Response to comment 1:** We agree that well-studied tasks like survey generation also involve extracting insights from papers. However, beyond extraction, ResearcherBench further focuses on the system's ability to reason over and integrate information from diverse sources to generate novel insights. Our evaluation framework is specifically designed to measure this higher-order capability. We believe this transition — from "information aggregator" to "analytical synthesizer" — is the defining characteristic of a genuine AI research partner, as opposed to a traditional summarization tool. **We appreciate the reviewer's attention to this nuance, and we have revised the Abstract to clarify this distinction.**
>
> **Comment 2 (Novelty of the evaluation framework):** The proposed evaluation pipeline using expert-curated criteria is reasonable, and I appreciate the annotation efforts. However, it appears to be similar to evaluation methods based on manually created references [1], and its advantages are unclear.
>
> **Response to Comment 2:** Thank reviewer for recognizing our annotation efforts. We would like to clarify the distinct necessity of our approach by comparing the methodologies and task natures of the two benchmarks.
>
> Firstly, DeepResearch Bench [1] typically operates by establishing top-level dimensions (Comprehensiveness, Insight/Depth, etc.) and using a Judge LLM to generate task-specific weights and criteria, calculating scores by comparing model outputs against manually created references. In contrast, ResearcherBench employs a rigorous three-stage method (Section 4.1.1) where human experts — not LLMs — manually design the specific rubrics. We score DARS based on whether their responses cover these expert-defined criteria.
>
> The necessity of our approach stems from the fundamental difference in the type of questions asked: DeepResearch Bench focuses largely on **information retrieval questions** and factual synthesis (e.g., "From 2020 to 2050, how many elderly people will there be in Japan? ..." [2]). For these determinate questions, LLM-generated criteria are generally reliable and verifiable. While ResearcherBench focuses on **frontier AI research questions** (e.g., "How can research on an agent's planning capabilities, as well as an AI's understanding and simulation of the real world... be systematically approached?"). Because these frontier problems lack a standard "ground truth" and require high-level conceptual reasoning, current LLMs cannot reliably generate valid evaluation dimensions on their own. Therefore, systematic investigation and rubric design by human domain experts is not just "reasonable" but essential for a valid evaluation.
>
> It should be clarified that we aim not to rank one benchmark over the other, but just to highlight that the expert-curated rubric in ResearcherBench are strictly necessary to address the unique challenges of evaluating such open-ended, frontier AI research questions.
>
> **Comment 3 (Rationale behind the dataset curation):** The authors claim to target "frontier AI research questions," but they do not systematically explain how the research questions in the introduced dataset are similar to or different from those in previous studies.
>
> Response to Comment 3: We have detailed our dataset construction methodology and its distinctiveness in Section 3 and Appendix B. Unlike traditional Research Task Benchmarks [3,4,5], which typically filter questions from massive databases of existing journal or conference papers (where the answers are already established), we curated questions from three authentic, frontier scenarios: **laboratory research discussions, interviews with leading AI researchers, and scientific forum discussions**. This sourcing strategy ensures that our questions target genuinely unsolved, high-value problems at the cutting edge of the field, rather than checking for retrospective knowledge. Furthermore, these sources allow us to capture rich contextual information, which is essential for our experts to design valid and comprehensive rubrics.

---

> > ### Comment · Reviewer_oGgf · 2025-11-21
> >
> > Thank you for your response. Since you claim an advantage for the proposed evaluation framework (Comment 2), I consider it necessary for the paper to experimentally demonstrate that the proposed approach leads to improvements in LLM-based evaluation. I would be willing to increase my score once this concern has been addressed.
> >
> > **Comment 2**
> >
> > I agree that your evaluation framework, which incorporates expert-annotated criteria and weights, is conceptually reasonable. However, the paper does not provide experimental justification that the proposed approach is superior to alternative methods that may require lower annotation costs.
> >
> > Specifically, since you claim that "current LLMs cannot reliably generate valid evaluation dimensions on their own," the paper should include an experimental comparison between this baseline and your proposed approach to demonstrate that the latter yields improved performance (e.g., in terms of alignment with human evaluation).
> >
> > **Comment 3**
> >
> > Thank you for the clarification. I now understand what “frontier” means in this context.

---

> > > ### Author Response · Authors · 2025-11-21
> > > **Official Comment by Authors**
> > >
> > > **Response to Comment 2 (Evaluation Framework Validation):**
> > >
> > > We sincerely thank the reviewer for this critical suggestion and for the willingness to reconsider the score. We are immediately initiating this experiment to compare our "Human-Expert Rubric" approach against an "LLM-Generated Rubric" baseline. We will report these findings as soon as the experiment is concluded.
> > >
> > > **Response to Comment 3:** Thanks for reviewer's confirmation. Glad to see our clarification was helpful !

---

> > > ### Author Response · Authors · 2025-11-28
> > > **Response to Comment 2: Comparison with LLM-Generated Baselines**
> > >
> > > Thanks for reviewer’s valuable comment. **We have supplemented a comprehensive comparative study in Appendix G, "Human-Designed vs. LLM-Generated Rubrics: A Comparative Analysis"**, to provide experimental justification that expert annotation demonstrates its superiority over automated LLM-generated methods.
> > >
> > > In Section G.1, we designed two distinct LLM-generated baselines to represent common automated evaluation strategies:
> > > 1. $\text{Rubric}_\text{dim}$ : Rubrics generated based on four generic dimensions (Comprehensiveness, Insight, Instruction-Following, and Readability), following the approach used in DeepResearch Bench [1].
> > > 2. $\text{Rubric}_\text{ref}$ : Rubrics constructed using our proposed three-stage method but substituting human experts with Claude-3.7-Sonnet during the *Criteria Design* and *Quality Control* phases, with the same reference materials.
> > >
> > > We evaluated all responses of DARS and LLMs with search tool, using these baselines and conducted a multi-dimensional analysis (including Coverage Score Analysis in G.2, Human Alignment in G.3, Weight Distribution in G.4, and Case Studies in G.5). Our key findings are as follows:
> > >
> > > - **LLM-generated rubrics lack discriminative power or alignment:**
> > >
> > > **$\text{Rubric}_\text{dim}$ exhibits severe ceiling effects.** As shown in Table 8, all systems achieved extremely high scores (0.87 ~ 0.98) with minimal variance using this rubric, indicating that generic criteria based on several predefined dimensions are excessively lenient and fail to distinguish model capabilities.
> > >
> > > **$\text{Rubric}_\text{ref}$ is overly rigid and misaligned.** While it offers better differentiation, it suffers from low Recall (0.60) compared to human experts. As detailed in Section G.3 and Table 9, LLMs tend to generate overly strict criteria based on specific keywords in reference materials, penalizing other valid but alternative approaches.
> > >
> > > - **Weight distribution reveals fundamental limitations of LLMs.** In Section G.4, we analyzed the importance weights assigned to criteria. We found that LLM-generated rubrics are dominated by "nice-to-have" (weight=1) criteria (73.66%), with a severe scarcity of "essential" (weight=3) criteria (8.93%). This confirms that without domain expertise, LLMs struggle to identify what is truly critical for answering frontier AI research questions.
> > >
> > > - **Mitigating risks of "Circular Dependency" and "Benchmark Contamination":** A critical concern in AI evaluation is whether using LLMs in the loop leads to **"light editing"** of model outputs or measures **"LLM-to-LLM imitation fidelity"** rather than independent research capability. Our experiment definitively refutes this. The significant divergence in content, weighting, and human alignment (F1=0.80 for Expert vs. F1=0.70 for $\text{Rubric}_\text{ref}$) demonstrates that **human experts contribute original, high-value reasoning** that fundamentally differs from the LLM's epistemic biases. The expert-designed rubric is not merely a subset of LLM outputs but a distinct, higher-quality standard.
> > >
> > > - **Expert-Designed rubrics achieve optimal human alignment**: Our proposed method demonstrates the best trade-off, achieving a balanced F1-score of 0.80 and substantial agreement with human expert judgments.
> > >
> > > These results empirically justify that while expert annotation incurs higher costs, it is currently indispensable for ensuring a valid and reliable evaluation in the domain of frontier AI research.
> > >
> > > ---
> > >
> > > [1] Du, Mingxuan, Benfeng Xu, Chiwei Zhu, Xiaorui Wang and Zhendong Mao. DeepResearch Bench: A Comprehensive Benchmark for Deep Research Agents. arXiv:2506.11763, arXiv, 2025.

---

> ### Author Response · Authors · 2025-11-20
> **Official Comment by Authors**
>
> **Comment 4 (Dataset size):** The introduced dataset includes only 65 research questions. While I understand the high cost of dataset creation due to expert annotation, the paper should still justify that a meaningful evaluation can be achieved with only 65 questions.
>
> **Response to Comment 4:** We thank the reviewer for acknowledging the high cost associated with expert annotation. As detailed in Section 3.1, our final dataset was distilled from an initial corpus of 932 candidate questions through a rigorous filtering process (Section 3.2) to 65 high-quality questions. These span 35 distinct AI research areas, ranging from model architecture to AI ethics. We prioritized quality over quantity to ensure each question represents a genuine frontier challenge. We believe this high density of expert adjudication and comprehensive topic coverage sufficiently compensates for the dataset size.
>
> Empirically, results in Table 3 also confirm the dataset's high discriminative power despite its size. It successfully stratifies performance, sharply distinguishing dedicated DARS from baselines (e.g., OpenAI at 0.7032 vs. GPT-4o Search at 0.3576) and revealing significant gaps within DARS implementations (e.g., vs. Grok3 family at ~0.44). This indicates that the dataset provides a high signal-to-noise ratio effective for differentiating frontier capabilities.
>
> Furthermore, as outlined in Appendix A.2 (Future Works), we have committed to Continuous Benchmark Evolution, where we will regularly incorporate new research questions that reflect the latest developments in AI.
>
> **Comment 5 (Missing human performance):** Including human performance in Table 2 would be helpful to assess how capable existing systems are for each task.
>
> **Response to Comment 5:** We appreciate this valuable suggestion. To address this, we have recruited human annotators and are conducting a human performance evaluation to establish a meaningful baseline for our benchmark. We anticipate releasing the result within one week to provide a clearer reference.
>
> **Comment 6:** I recommend using \citep when suitable.
>
> **Response to Comment 6:** We have updated the citation format in the revised paper, adopting \citep where appropriate to improve readability. Thank for reviewer’s attention to detail.
>
> **We sincerely appreciate the reviewer's constructive suggestions and believe that the additional explanations significantly improve the quality of our submission. We hope that this provides sufficient reasons to raise the score.**
>
> ---
>
> [1] Du, Mingxuan, Benfeng Xu, Chiwei Zhu, Xiaorui Wang和Zhendong Mao. DeepResearch Bench: A Comprehensive Benchmark for Deep Research Agents. arXiv:2506.11763, arXiv, 2025.
>
> [2] https://huggingface.co/datasets/muset-ai/DeepResearch-Bench-Dataset
>
> [3] Asai, Akari, Jacqueline He, Rulin Shao, et al. OpenScholar: Synthesizing Scientific Literature with Retrieval-Augmented LMs. arXiv:2411.14199, arXiv, 2024.
>
> [4] Yifei, Li S., Allen Chang, Chaitanya Malaviya and Mark Yatskar. ResearchQA: Evaluating Scholarly Question Answering at Scale Across 75 Fields with Survey-Mined Questions and Rubrics. arXiv:2509.00496, arXiv, 2025.
>
> [5] Liu, Yujie, Zonglin Yang, Tong Xie, et al. ResearchBench: Benchmarking LLMs in Scientific Discovery via Inspiration-Based Task Decomposition. arXiv:2503.21248, arXiv, 2025.

---

> > ### Comment · Reviewer_oGgf · 2025-11-21
> >
> > Thank you for your response. The points I listed under Minor Limitations did not significantly influence my final score, so you may prioritize other experiments. I would like to offer the following comments as suggestions for further improving the paper, and I would appreciate it if you could address them before the final version is published.
> >
> > > It successfully stratifies performance, sharply distinguishing dedicated DARS from baselines (e.g., OpenAI at 0.7032 vs. GPT-4o Search at 0.3576) and revealing significant gaps within DARS implementations (e.g., vs. Grok3 family at ~0.44).
> >
> > To substantiate this claim with a small dataset, I recommend providing statistical information such as confidence intervals. You may find the following papers useful as references:
> >
> > * https://aclanthology.org/W04-3250/
> > * https://aclanthology.org/2025.emnlp-main.1025/ (Table 2)

---

> > > ### Author Response · Authors · 2025-11-21
> > > **Official Comment by Authors**
> > >
> > > **Response to the Follow-up Comment:**
> > >
> > > We sincerely thank the reviewer for the constructive feedback and the valuable references. We will proceed with these experiments, to calculate confidence intervals for the results in Table 3 immediately. We commit to incorporating this statistical analysis into the final version of the paper to further strengthen the reliability of our evaluation.

---

> > > ### Author Response · Authors · 2025-11-28
> > > **Response to Comment: Confidence Intervals**
> > >
> > > Thanks for reviewer’s constructive suggestion to enhance the statistical rigor of our evaluation.
> > >
> > > In **Section 5.2 (Main Results)** of the revised paper, we have calculated and reported the **95% confidence intervals** for the Rubric Assessment Coverage scores. Following the methodology described by Muennighoff et al. (2025) [1], we performed **10,000 bootstrap resampling iterations** on our evaluation results.
> > >
> > > The statistics are presented in the Table 3 below:
> > >
> > > | Model | Rubric Assessment Coverage (with 95% CI) |
> > > | :--- | :--- |
> > > | **OpenAI Deep Research** | **0.7032** `[-0.050, +0.049]` |
> > > | Gemini Deep Research | 0.6929 `[-0.053, +0.052]` |
> > > | Claude Research | 0.6113 `[-0.053, +0.053]` |
> > > | Mita Deep Research | 0.5835 `[-0.057, +0.059]` |
> > > | Doubao Deep Research | 0.5754 `[-0.057, +0.055]` |
> > > | Perplexity Deep Research | 0.4800 `[-0.056, +0.055]` |
> > > | Grok3 DeepSearch | 0.4414 `[-0.054, +0.057]` |
> > > | Grok3 DeeperSearch | 0.4398 `[-0.056, +0.056]` |
> > > | Perplexity: Sonar Reasoning Pro | 0.4663 `[-0.056, +0.056]` |
> > > | GPT-4o Search Preview | 0.3576 `[-0.051, +0.052]` |
> > >
> > > As shown in the table, the confidence intervals are generally contained within a range of $\pm 0.05$ approximately. We believe that these margins provide a reasonable degree of reliability. Specifically, the confidence intervals support the distinct performance stratification observed in our experiments, clearly distinguishing top-tier DARS (e.g., OpenAI at $0.70 \pm 0.051$, Gemini at $0.69 \pm 0.052$) from other baselines (e.g., GPT-4o at $0.35 \pm 0.051$, or Grok3 family at $0.44 \pm 0.055$). It suggests that the performance gaps revealed by our benchmark are statistically meaningful.
> > >
> > > We hope this additional statistical analysis can substantiate our claims and address reviewer’s concerns regarding statistical validation. **If these additional verifications could improve the quality of our submission, we would be grateful if reviewer would consider raising the score.**
> > >
> > > ---
> > >
> > > [1] Muennighoff, Niklas, et al. "S1: Simple Test-Time Scaling." arXiv preprint arXiv:2501.19393 (2025).

---

### Official Review · Reviewer_VdGR · 2025-10-31

**Soundness:** 3
**Presentation:** 3
**Contribution:** 3
**Rating:** 6
**Confidence:** 2

**Summary:**

The paper introduces ResearcherBench, a benchmark for evaluating advanced AI research systems (DARS) on real-world AI research questions. It includes tasks across 35 AI subfields and uses both rubric-based and factual assessments to measure reasoning depth, factual accuracy, and citation quality. Results show that systems like OpenAI Deep Research and Gemini perform well in generating insights but struggle with technical depth and reproducibility, highlighting current limits of AI as true research collaborators.

**Strengths:**

- The paper focuses on rapidly emerging class of AI tools designed to assist in real research tasks, moving beyond traditional QA or chat systems.
- ResearcherBench is built from real research problems collected from expert interviews, lab discussions, and research forums, rather than synthetic or exam-style datasets. It closely reflects real-world scientific inquiry.
- The study shows current systems perform well in generating insights and high-level reasoning but struggle with deep technical or reproducible details, providing valuable direction for future AI research assistants.

**Weaknesses:**

- The benchmark focuses solely on AI research questions, so generalization to other scientific disciplines remains unknown.
- The evaluation measures single-turn performance, not long-term or iterative research interactions that mimic real collaboration between humans and AI.
- The framework evaluates the quality of final answers, but not how systems search, reason, or decide what to include, limiting interpretability of model behavior.

**Questions:**

- How would the systems perform in long-term, multi-turn research collaborations instead of single-turn evaluations?
- How to measure how systems actually search, think, and make decisions along the way?

---

> ### Author Response · Authors · 2025-11-20
> **Official Comment by Authors**
>
> Thanks for reviewer's comments. Here are our responses to the comments.
>
> **Comment 1:** The benchmark focuses solely on AI research questions, so generalization to other scientific disciplines remains unknown.
>
> **Response to Comment 1:** We acknowledge this limitation. Our decision to focus exclusively on AI research questions stems from our team's specific domain expertise. As stated in Appendix A.1 (Limitations), selecting truly "frontier" research questions and designing rigorous, expert-level rubrics requires deep, specialized knowledge. We believe that expanding into other disciplines (e.g., physics, chemistry) without corresponding domain expertise would compromise the validity of the benchmark. As discussed in Appendix A.2 (Future Works), we have explicitly identified "Cross-Domain Expansion" as a critical direction. We plan to collaborate with experts from other scientific fields to expand our dataset and validate our conclusions in future iterations.
>
> **Comment 2 & Question 1:** The evaluation measures single-turn performance, not long-term or iterative research interactions that mimic real collaboration between humans and AI. How would the systems perform in long-term, multi-turn research collaborations instead of single-turn evaluations?
>
> **Response to comment 2 & Question 1:** The current generation of leading commercial Deep AI Research Systems (DARS) evaluated in our work primarily operates as **single-turn, "black-box" systems**. Users typically provide an initial prompt, and the system autonomously executes the workflow to generate a final report without intermediate human interaction. Consequently, it is currently not feasible to evaluate long-term or iterative human-AI collaboration using these commercial web UIs. As we continue to maintain ResearcherBench, we intend to expand our evaluation framework to include interactive and longitudinal collaboration settings as soon as DARS with these capabilities become widely available, as noted in our Future Works.
>
> **Comment 3 & Question 2:** The framework evaluates the quality of final answers, but not how systems search, reason, or decide what to include, limiting interpretability of model behavior. How to measure how systems actually search, think, and make decisions along the way?
>
> **Response to Comment 3 & Question 2:** Similar to our response above, the "black-box" nature of commercial DARS poses a significant challenge for process-level evaluation. While some systems (e.g., Gemini Deep Research, Claude Research) display partial reasoning traces, there is currently no standardized framework for exposing search queries, reasoning steps, or decision-making logic across different DARS providers. We are closely monitoring the development of open-source DARS and architectures, which will allow for the fine-grained process evaluation the reviewer suggests in future iterations of our work.

---

### Official Review · Reviewer_Awa7 · 2025-10-31

**Soundness:** 3
**Presentation:** 4
**Contribution:** 3
**Rating:** 8
**Confidence:** 4

**Summary:**

This paper introduces ResearcherBench, a new benchmark designed to evaluate Deep AI Research Systems on frontier AI research questions. The benchmark includes 65 carefully curated questions across technical details, literature review, and open-ended consulting, paired with a dual evaluation framework that assesses both insight quality and factual grounding of leading LLMs.

**Strengths:**

1, The paper assesses whether deep-research systems can provide meaningful insight and consultation on unsolved AI research problems; this is a timely and interesting problem as AI4Science is a rising trend;

2, It provides a new dataset;

3, Results show some emergent strengths and weaknesses not captured by prior benchmarks.

**Weaknesses:**

1, A notable limitation of this work lies in its evaluation methodology for frontier research questions. Although the benchmark emphasizes unsolved and open-ended research problems, the scoring relies primarily on fixed rubrics that reward coverage of predefined expert criteria. This is less reliable for those open, unsolved problems. A better way could be incorporating comparative or pairwise evaluation frameworks (e.g., tournament-style or blind ranking) that assess relative research merit and originality rather than absolute rubric compliance.
2, The evaluation does not capture whether the model produces surprising or genuinely helpful insights even though it claims overall helpful, but when and how---this is unclear. For frontier research tasks, simply covering known points is insufficient, as recent LLMs are already strong at that. I would suggest asking annotators to mark which segments actually advanced their thinking, and give more fine-grained analysis to discover the essential value of LLMs towards humans.

**Questions:**

N/A

---

> ### Author Response · Authors · 2025-11-20
> **Official Comment by Authors**
>
> Thanks for reviewer's comments. Here are our responses to the comments.
>
> **Comment 1:** A notable limitation of this work lies in its evaluation methodology for frontier research questions. Although the benchmark emphasizes unsolved and open-ended research problems, the scoring relies primarily on fixed rubrics that reward coverage of predefined expert criteria. This is less reliable for those open, unsolved problems. A better way could be incorporating comparative or pairwise evaluation frameworks (e.g., tournament-style or blind ranking) that assess relative research merit and originality rather than absolute rubric compliance.
>
> **Response to Comment 1:** For the Rubric Assessment, our design does not require DARS to match a specific "ground truth." Instead, it aims to identify valuable dimensions of reasoning and insight. This approach allows models to explore freely and demonstrate capabilities within these dimensions, rather than prescribing fixed answers or demanding coverage of predefined solutions.
>
> Please refer to Example 2 in Appendix C.2 of the revised paper. Consider the question: "How should the development of generative AI evolve: focusing on dialogue-based systems (Chat) or autonomous action-taking systems (Agent)?" Our rubric evaluates the identification of valuable insight dimensions, such as:
> - "Provides relevant historical analogies to similar technological evolutions (e.g., smart speakers, voice assistants)" (Weight=1).
>
> The key is that the rubric rewards the demonstration of such historical analogies, regardless of the specific analogy used. This design acknowledges that frontier research lacks fixed answers. **The rubric acts as a guideline for identifying insight quality, not as a rigid checklist of facts.**
>
> Regarding the suggestion to incorporate **pairwise evaluation frameworks**, we actually conducted preliminary experiments with this method. However, our analysis revealed a severe length bias in pairwise comparisons (Pearson correlation between win rate and response length: r=0.9487, p<0.001), rendering it unsuitable as a primary evaluation metric. In contrast, our rubric-based assessment effectively mitigates this bias through binary judgment (satisfied/not satisfied) of specific criteria. **We have added detailed experimental results regarding pairwise comparisons to Appendix E.7 of the revised paper. Thanks for reviewer’s insightful suggestion.**
>
> **Comment 2:** The evaluation does not capture whether the model produces surprising or genuinely helpful insights even though it claims overall helpful, but when and how---this is unclear. For frontier research tasks, simply covering known points is insufficient, as recent LLMs are already strong at that. I would suggest asking annotators to mark which segments actually advanced their thinking, and give more fine-grained analysis to discover the essential value of LLMs towards humans.
>
> **Response to Comment 2:** We fully agree with the reviewer that Deep AI Research Systems (DARS) should be evaluated on their ability to generate genuinely helpful insights. To ensure our benchmark captures this ability, our human experts conducted extensive research on each frontier AI question to construct rubrics based on deep, expert-level insights rather than superficial facts. Therefore, we believe that a high score on ResearcherBench is strong evidence of a system's ability to provide genuinely helpful insights that align with expert consensus on frontier topics.
>
> This is also empirically supported by our results, which demonstrate that standard LLMs with search engines cannot easily pass the benchmark by simply covering basic points. As shown in Table 3, GPT-4o Search Preview achieved a rubric coverage score of only 0.3576, significantly lower than OpenAI Deep Research (0.7032). This substantial performance gap confirms that our benchmark effectively distinguishes between the basic information retrieval capabilities of general LLMs and the advanced insight synthesis capabilities required of DARS, proving that "simply covering known points" is insufficient for a high score.
>
> Regarding the reviewer's suggestion to "ask annotators to mark which segments actually advanced their thinking," we acknowledge this as a very valuable perspective for user-centric evaluation. However, we opted not to include this evaluation in the current version, due to the high subjectivity and potential bias inherent in individual annotators' backgrounds and cognitive contexts. We prioritized a standardized and reproducible evaluation framework to ensure fairness across different systems.

---

> > ### Comment · Reviewer_Awa7 · 2025-11-25
> > **Author rebuttal acknowledged**
> >
> > Author rebuttal acknowledged

---

### Official Review · Reviewer_msmb · 2025-10-31

**Soundness:** 2
**Presentation:** 2
**Contribution:** 2
**Rating:** 2
**Confidence:** 4

**Summary:**

This paper introduces ResearcherBench, a new benchmark aimed at evaluating Deep AI Research Systems (DARS)—agentic systems designed to perform open-ended research tasks—on frontier AI research questions. The benchmark comprises 65 questions across 35 AI subfields, categorized into technical details, literature review, and open consulting. The authors design a dual evaluation framework combining: (1) A rubric assessment (human-designed, multi-criteria coverage scoring), and (2) A factual assessment (citation faithfulness and groundedness metrics). Several commercial DARS systems (e.g., OpenAI, Gemini, Claude, Grok, Perplexity) are evaluated using this framework. The paper concludes that DARS perform best on open-ended consulting tasks but struggle with technical detail and literature review questions. The benchmark is open-sourced for community use.

**Strengths:**

1. The paper clearly identifies a genuine gap in current DARS evaluation: most existing benchmarks focus on retrieval and summarization rather than assessing insight generation for frontier research tasks.

2. Questions are sourced from real-world research scenarios (labs, interviews, forums), with multiple annotators.

3. The rubric + factual framework offers a complementary perspective on reasoning quality and factual reliability, which is more nuanced than single-score metrics.

**Weaknesses:**

1. The rubric coverage relies on automated scoring (o3-mini model), but the reliability of using LLM-as-a-Judge (even with an F1 of ~0.80) for nuanced insight evaluation is questionable. These tasks are inherently subjective.

2. Despite claiming expert input, many steps (insight extraction, initial drafting) depend on Claude-3.7-Sonnet, raising questions about the human-likeness of the benchmark itself.

3. There is no granular analysis of which topics or difficulty levels cause the most failure, which limits the interpretability of the evaluation outcomes.

4. The final results are judged entirely by automated systems. Without any qualitative human review of model answers, it is difficult to trust that the benchmark reflects actual scientific reasoning quality.

5. Although the paper frames itself as evaluating “frontier research”, it is limited entirely to AI. Generalizing conclusions about research assistance more broadly is premature.

6. A core methodological concern is the circular dependency between benchmark creation and evaluation. The benchmark positions itself as measuring whether DARS can behave like “expert researchers” — yet the supposed expert rubric and insight ground‑truths are themselves partially LLM‑generated (Claude‑3.7‑Sonnet performs insight extraction and drafts core rubric content). While humans later “review” and “refine,” it is unclear how much original human reasoning exists versus light editing of LLM output. This structure blurs the line between human‑defined truth and LLM‑defined norm, potentially baking the model’s epistemic biases into the gold standard it is later evaluated against. This risk is amplified by two additional problems:
- Benchmark contamination risk: Since modern frontier models share training data and emergent reasoning patterns, having one LLM generate key ideas and rubric criteria and another LLM evaluate performance risks epistemic contamination. The evaluation pipeline implicitly rewards models that think like Claude (or like the judging model), not models that demonstrate genuinely independent scientific reasoning. This undermines claims of measuring researcher‑like capability — instead, the benchmark may be measuring LLM‑to‑LLM imitation fidelity.
- Unclear grounding of “correctness” for open‑ended research: The paper treats extracted “key insights” and rubrics as if they represent human‑agreed, authoritative ground truth. But frontier research questions — by definition — do not have fixed answers. Alternative viewpoints, emerging results, or novel hypotheses could be equally valid yet score poorly because they diverge from the LLM‑generated rubric. This raises a deeper epistemic issue: If the benchmark itself cannot guarantee the correctness or completeness of its ground‑truth insight set, what exactly is being measured? The factual assessment further assumes that the model must cite a particular set of sources, even though real research often draws on multiple valid references, emerging conversations, or unpublished expert knowledge. Penalizing models for providing facts that deviate from the benchmark’s curated context risks enforcing narrow compliance rather than genuine reasoning ability.

**Questions:**

1. How much of the rubric and insight generation process is human-authored versus LLM-generated?
While you mention expert review and refinement, the initial insight extraction and rubric drafting are performed by Claude-3.7-Sonnet. Can you clarify what percentage of the rubric criteria and wording was contributed by humans versus accepted from the model? Was any rubric content written independently of LLM suggestions?

2. How do you mitigate contamination from using one LLM to generate benchmark expectations and another LLM to evaluate responses?
Given that LLMs often share training data and reasoning styles, how can you be confident the benchmark is measuring generalizable research ability rather than model-to-model imitation? For example, wouldn’t Claude-like answers be favored in rubric alignment and factual assessments?

3. Why are open-ended research answers evaluated against fixed rubrics and citation sets?
In many research contexts, multiple valid answers and citation paths exist. If a model provides a novel or expert-justified answer that diverges from the expected rubric or cites a different source, how is this handled? Is there a mechanism to avoid penalizing correct but non-aligned outputs?

4. Was there any human evaluation of DARS outputs?
Aside from the small meta-evaluation for judge model selection, were any model responses reviewed by domain experts to validate whether rubric and factual scores match human intuitions about insight quality and correctness?

5. Why not provide per-subject or per-difficulty breakdowns of system failures?
A more detailed error analysis (e.g., which topics consistently trip up models) would help validate the benchmark’s utility and reveal meaningful performance patterns. Was this considered?

6. Can you justify generalizing your conclusions beyond AI research?
The paper implies that DARS are becoming viable partners for “frontier research” in general, yet your entire benchmark is constrained to AI. How should readers interpret this claim without cross-domain evidence?

---

> ### Author Response · Authors · 2025-11-20
> **Official Comment by Authors**
>
> Thanks for reviewer's comments. Here are our responses to the comments.
>
> **Comment 1:** The rubric coverage relies on automated scoring (o3-mini model), but the reliability of using LLM-as-a-Judge (even with an F1 of ~0.80) for nuanced insight evaluation is questionable. These tasks are inherently subjective.
>
> **Response to Comment 1:** We acknowledge that evaluating nuanced insights is challenging, however, we believe our approach is robust for the following reasons:
>
> 1) We ground our metric reliability on the established paper proposed by Landis & Koch (1977) [1]. According to their scale, agreement coefficients between 0.61 and 0.80 are classified as "Substantial," while 0.81–1.00 are "Almost Perfect." Our experiments yield a Human-Human Agreement (Cohen’s Kappa) of κ=0.67 and a Human-LLM Agreement (F1-score) of 0.80. Both metrics fall within the "Substantial" range, providing an empirical evidence for the reliability of our judge model.
>
> 2) Our alignment scores are comparable to other recent high-quality benchmarks that utilize LLM-as-a-Judge (e.g., PaperBench reports an F1-score of 0.83) [2,3,4]. The consistency of our results (F1-score = 0.80) with these concurrent works further validates that our alignment evaluation performs at a standard acceptable to the research community.
>
> 3) Unlike open-ended quality scoring (e.g., measuring a score from 1 to 10), our rubric assessment decomposes complex evaluation into specific binary judgment tasks (i.e., determining whether a criterion is "satisfied" or "not satisfied"). This criteria-based approach explicitly minimizes subjectivity by only checking for the presence of specific content rather than making holistic value judgments.
>
>
> In conclusion, we consider the use of o3-mini for rubric assessment to be both reasonable and reliable within our framework. Based on reviewer’s comment, we have revised "Section 4.3 Meta Evaluation And Judge Model Selection" ("Section 4.3 Judge Model Selection Via Human Evaluation" in the revised paper) to better highlight these reliability metrics and the rationale for our methodology. Thanks for reviewer’s suggestions.
>
> **Comment 2:** Despite claiming expert input, many steps (insight extraction, initial drafting) depend on Claude-3.7-Sonnet, raising questions about the human-likeness of the benchmark itself.
>
> **Response to comment 2:**  We would like to clarify that **the evaluation rubrics were independently generated by human experts, and the LLM-generated content served solely as auxiliary reference material during the annotation process; it does not appear in the final rubrics.**
>
> We have detailed the workflow of rubric construction in Section 4.1.1. Specifically, we only utilized LLM in Stage 1: Insight Extraction to assist human annotators in collecting and organizing information. As stated in the paper, these generated comprehensive reference materials only "serve as auxiliary information" Besides, we also required human annotators to review, validate, and supplement these materials to ensure their reliability.
>
> In Stage 2: Criteria Design, human experts used these auxiliary materials to familiarize themselves with the research tasks, take key insights as references and design the specific rubric criteria independently. Furthermore, we implemented a multi-stage quality control process in Stage 3 to guarantee the reliability and validity of our final rubrics. In conclusion, rubrics are written by human expert annotators, with no LLM-generated draft version.
>
> Based on reviewer’s comments, we have revised the descriptions in Section 4.1.1 (Rubric Construction) and the corresponding sections in the Appendix C of the revised paper to clarify these details and prevent potential misunderstandings. Thanks for reviewer’s suggestions.
>
> **Comment 3:** There is no granular analysis of which topics or difficulty levels cause the most failure, which limits the interpretability of the evaluation outcomes.
>
> **Response to comment 3:** Appreciate the reviewer’s valuable suggestion. To address the concern regarding granular analysis, we have included **a detailed Error Analysis in Appendix E.6 of the revised paper**. This section provides a granular breakdown of failure patterns and their underlying causes. For example, our analysis reveals that errors often stem from relevant information not being retrieved during the search phase (leading to "Incompleteness" error). Besides, we observed that failure modes in factual assessment vary significantly across different Deep AI Research Systems (DARS) designs (e.g., distinguishing between invalid citations and unsupported hallucinations). We believe this addition enhances the interpretability of our evaluation outcomes.

---

> ### Author Response · Authors · 2025-11-20
> **Official Comment by Authors**
>
> **Comment 4:** The final results are judged entirely by automated systems. Without any qualitative human review of model answers, it is difficult to trust that the benchmark reflects actual scientific reasoning quality.
>
> **Response to comment 4:** As noted in **"Response to Comment 1"**, we conducted extensive Human Evaluation experiments to establish the reliability of our automated evaluation system in the Section 4.3. By systematically comparing automated evaluation results against human expert annotations as ground-truth, we demonstrated that our selected judge model achieves high alignment with human judgments (e.g., F1-score = 0.80). In the revised paper, we have refined the descriptions in Section 4.3 to provide a clearer explanation of our human evaluation methodology and how it ensures the validity of the benchmark results.
>
> **Comment 5:** Although the paper frames itself as evaluating “frontier research”, it is limited entirely to AI. Generalizing conclusions about research assistance more broadly is premature.
>
> **Response to comment 5:** We respectfully clarify that our work never claims to evaluate **"frontier research"** in a general sense. On the contrary, we have consistently defined and emphasized our scope as **"frontier AI Research"** throughout the paper (e.g., in the Title and Introduction).
>
> We acknowledge that generalizing conclusions to other scientific fields is a significant challenge. However, given that our team’s expertise lies primarily within the AI domain, we exercised caution regarding non-AI disciplines (such as physics and chemistry). We believe that curation of representative frontier research questions and the design of rigorous rubrics for these fields require specialized domain knowledge that we currently do not possess. We fully agree with reviewer’s comment and recognize the importance of broader applicability. As discussed in Appendix A.2 (Future Works), we have explicitly identified "Cross-Domain Expansion" as a critical direction. We plan to collaborate with experts from other scientific fields to expand our dataset and validate our conclusions in future iterations.
>
> **Comment 6:** A core methodological concern is the circular dependency between benchmark creation and evaluation. The benchmark positions itself as measuring whether DARS can behave like “expert researchers” — yet the supposed expert rubric and insight ground‑truths are themselves partially LLM‑generated (Claude‑3.7‑Sonnet performs insight extraction and drafts core rubric content). While humans later “review” and “refine,” it is unclear how much original human reasoning exists versus light editing of LLM output. This structure blurs the line between human‑defined truth and LLM‑defined norm, potentially baking the model’s epistemic biases into the gold standard it is later evaluated against. This risk is amplified by two additional problems:
>
> **Response to comment 6:** We wish to clarify the methodology regarding the use of LLM-generated materials to address concerns about circular dependency.
>
> **Firstly, the insight ground-truths are not generated by the LLM.** They originate from the massive research context we collected from authentic scientific scenarios (e.g., laboratory discussions and expert interviews). The role of LLM in Stage 1 is strictly limited to extracting these pre-existing key insights from the context. Furthermore, this process undergoes rigorous human review, validation, and supplement to ensure the accuracy and comprehensiveness of the background understanding.
>
> **Secondly, as detailed in Stage 2: Criteria Design, the LLM-generated reference materials only "serve as auxiliary materials".** Human experts leverage these materials to familiarize themselves with the specific research context, and they are required to design the rubric criteria by their own, taking extracted key insights as reference, instead of copying them directly, based on Annotation Guidelines in the Appendix C.1.2. The final rubrics are authored entirely by human annotators, and no LLM-generated content is directly included.
>
> Thirdly, the multi-stage quality control process in Stage 3 further reviews the rubrics to guarantee their reliability and validity.
>
> In conclusion, neither the expert-designed rubrics nor the insight ground-truths are LLM-generated; Claude-3.7-Sonnet serves merely as an assistant for preliminary information organization. We have revised the relevant sections in the paper to prevent similar misunderstandings for future readers. Thanks again for reviewer’s highlighting this importance of clarity.

---

> ### Author Response · Authors · 2025-11-20
> **Official Comment by Authors**
>
> **Comment 6.1 (Benchmark contamination risk):** Since modern frontier models share training data and emergent reasoning patterns, having one LLM generate key ideas and rubric criteria and another LLM evaluate performance risks epistemic contamination. The evaluation pipeline implicitly rewards models that think like Claude (or like the judging model), not models that demonstrate genuinely independent scientific reasoning. This undermines claims of measuring researcher ‑like capability — instead, the benchmark may be measuring LLM‑to‑LLM imitation fidelity.
>
> **Response to comment 6.1:** Based on **“Response to comment 6”**, we emphasize that the key insights evaluation criteria are human-written, and the key insights are derived from authentic research contexts. The evaluation pipeline does not rely on the model's internal epistemic biases, therefore "contamination risk" or measuring "LLM-to-LLM imitation fidelity" is effectively mitigated.
>
> **Comment 6.2 (Unclear grounding of “correctness” for open‑ended research):** The paper treats extracted “key insights” and rubrics as if they represent human‑agreed, authoritative ground truth. But frontier research questions — by definition — do not have fixed answers. Alternative viewpoints, emerging results, or novel hypotheses could be equally valid yet score poorly because they diverge from the LLM‑generated rubric. This raises a deeper epistemic issue: If the benchmark itself cannot guarantee the correctness or completeness of its ground‑truth insight set, what exactly is being measured? The factual assessment further assumes that the model must cite a particular set of sources, even though real research often draws on multiple valid references, emerging conversations, or unpublished expert knowledge. Penalizing models for providing facts that deviate from the benchmark’s curated context risks enforcing narrow compliance rather than genuine reasoning ability.
>
> **Response to comment 6.2:**
>
> 1. **Clarification on Rubric Flexibility:** First, referring to **“Response to Comment 6”**, the rubrics are not LLM-generated. More importantly, regarding the epistemic concern, our rubric assessment does not demand a specific "ground truth." Instead, it aims to identify valuable dimensions of reasoning and insight. The framework allows DARS to explore freely within some high-level dimensions, rather than checking for compliance with pre-defined, specific answers.
>
>
> Please refer to Example 2 in Appendix C.2 of revised paper. Considering the question “How should the development of generative AI evolve: focusing on dialogue-based systems (Chat) or autonomous action-taking systems (Agent)? What are the key differences, technological requirements, and future implications of each approach”, Our rubric evaluates the presence of valuable insight dimensions, such as:
>  - "Provides relevant historical analogies to similar technological evolutions (e.g., smart speakers, voice assistants)" (Weight=1).
>
> The key is that the rubric rewards the demonstration of such historical analogies, regardless of the specific analogy used. This design acknowledges that frontier research lacks fixed answers. **The rubric acts as a guideline for identifying insight quality, not as a rigid checklist of facts.** Consequently, models will not be penalized for exploring diverse perspectives as long as they demonstrate the valuable dimensions required by the criteria.
>
> 2. **Clarification on Factual Assessment:** We respectfully clarify that our factual assessment does not assume that the DARS must cite a particular set of sources. Instead, it evaluates the response of DARS through Faithfulness and Groundedness: Faithfulness measures whether the claims in the response are actually supported by their URL sources (retrieved by DARS), while Groundedness measures the overall citation coverage of the response content. **Therefore, factual assessment focuses on whether the cited claims are factually supported by their references, not whether the model cites a specific "gold standard" bibliography.** We do not penalize models for utilizing valid references outside of our own context materials.
>
> We hope this clarifies how we define and measure "correctness" (or rather, research effectiveness) within ResearcherBench.

---

> ### Author Response · Authors · 2025-11-20
> **Official Comment by Authors**
>
> **Question 1:** How much of the rubric and insight generation process is human-authored versus LLM-generated? While you mention expert review and refinement, the initial insight extraction and rubric drafting are performed by Claude-3.7-Sonnet. Can you clarify what percentage of the rubric criteria and wording was contributed by humans versus accepted from the model? Was any rubric content written independently of LLM suggestions?
>
> **Response to question 1:**
>
> We respectfully clarify that for the Rubric Design, the specific criteria and wording are fully human-authored. It is important to distinguish between the role of the model in data processing versus standard definition:
>
> As mentioned in **“Response to comment 1”** and **“Response to comment 6”**, the model's role was strictly limited to processing the massive research context to extract key information in Stage 1. These generated summaries served solely as "auxiliary materials" to assist human experts in understanding the context efficiently.
>
> The actual drafting of the rubric — including the definition of criteria, specific wording, and weight assignment — was performed by human expert annotators. The humans took the extracted insights as reference material but did not "accept" rubric criteria from the model. Therefore, regarding the reviewer's question on "what percentage of the rubric criteria was contributed by humans," the answer is **100%**. For the phase of Insight Extraction, the model was primarily responsible for the initial workload. Human experts also participated by supplementing new insights, and the extent of this human contribution varied across different questions.
>
> **Question 2:** How do you mitigate contamination from using one LLM to generate benchmark expectations and another LLM to evaluate responses? Given that LLMs often share training data and reasoning styles, how can you be confident the benchmark is measuring generalizable research ability rather than model-to-model imitation? For example, wouldn’t Claude-like answers be favored in rubric alignment and factual assessments?
>
> **Response to question 2:** Please refer to **"Response to Comment 6"**. Since the rubrics are fully human-authored rather than generated by an LLM, the risk of contamination via "LLM-defined expectations" is effectively mitigated. Empirically, our main results in Section 5.2 directly contradict the hypothesis that the benchmark favors "Claude-like" answers. As shown in Table 3, Claude Research ranked third in the rubric assessment (Score: 0.6113), falling significantly behind Gemini Deep Research (Score: 0.6929) and OpenAI Deep Research (Score: 0.7032).
>
> **Question 3:** Why are open-ended research answers evaluated against fixed rubrics and citation sets? In many research contexts, multiple valid answers and citation paths exist. If a model provides a novel or expert-justified answer that diverges from the expected rubric or cites a different source, how is this handled? Is there a mechanism to avoid penalizing correct but non-aligned outputs?
>
> **Response to question 3:** Please refer to **"Response to Comment 6.2"**. Regarding the rubric assessment, our design aims to identify valuable dimensions of reasoning and insight rather than enforcing specific "ground-truth" answers. Regarding the factual assessment, our objective is not to restrict models to a fixed set of citation sources. Instead, we evaluate Faithfulness — verifying whether the claims extracted from the responses are factually supported by the citations URL, which the DARS itself retrieved. This ensures that models are rewarded for accurate self-grounding, regardless of which valid sources they choose to cite.

---

> ### Author Response · Authors · 2025-11-20
> **Official Comment by Authors**
>
> **Question 4:** Was there any human evaluation of DARS outputs? Aside from the small meta-evaluation for judge model selection, were any model responses reviewed by domain experts to validate whether rubric and factual scores match human intuitions about insight quality and correctness?
>
> **Response to question 4:** Human evaluation is primarily addressed in Section 4.3 (Judge Model Selection Via Human Evaluation) of the revised paper. To validate our evaluation framework, we recruited 5 domain expert annotators to evaluate a stratified subset of 20 responses using the identical rubrics employed in the rubric assessment. We treated these human expert judgments as the ground truth, to assess the alignment between the automated judge model and human preferences. The results demonstrated a substantial agreement between the judge model and human experts, achieving an F1 score of 0.80. It empirically validates the reliability of our LLM-as-a-Judge approach and confirms that the automated scores closely match human expert judgments.
>
> Regarding reviewer’s concern about "human intuitions about insight quality," while this is inherently difficult to quantify, we relied on qualitative monitoring during the annotation process. Notably, our expert annotators did not report any instances, where a model response appeared intuitively "excellent" but received a low score in the rubric assessment.
>
>
> **Question 5:** Why not provide per-subject or per-difficulty breakdowns of system failures? A more detailed error analysis (e.g., which topics consistently trip up models) would help validate the benchmark’s utility and reveal meaningful performance patterns. Was this considered?
>
> **Response to question 5:** We thank the reviewer for this constructive suggestion. In response, we have incorporated a comprehensive **Error Analysis in Appendix E.6 (specifically Table 6) of the revised paper**. This analysis breaks down failure modes across different question types and systems, revealing distinct performance patterns and causes. Our analysis reveals that Literature Review questions exhibit the highest rubric error rate (average 8.87 errors per response). This indicates that systematic coverage of research breadth remains a significant bottleneck, even for state-of-the-art DARS. We also found that failure patterns vary significantly depending on the system design. For instance, Mita Deep Research is more prone to errors caused by inaccessible URLs (Invalid Attribution), whereas Doubao Deep Research demonstrates relatively severe issues with hallucination problems (Unsupported Answers).
>
> We believe these granular insights effectively validate the benchmark’s utility in diagnosing specific weaknesses in current DARS architectures.
>
> **Question 6:** Can you justify generalizing your conclusions beyond AI research? The paper implies that DARS are becoming viable partners for “frontier research” in general, yet your entire benchmark is constrained to AI. How should readers interpret this claim without cross-domain evidence?
>
> **Response to question 6:**
>
> Please refer to our detailed explanation in **"Response to comment 5"**. We respectfully clarify that we have never claimed our benchmark evaluates "frontier research" in a general sense. On the contrary, we have consistently emphasized that our evaluation scope is strictly limited to "frontier AI Research", especially in Appendix A.1 (Limitation).
>
> Given that our team's expertise lies primarily within the AI domain, we maintain a cautious attitude towards generalizing our work to non-AI disciplines (such as physics or chemistry). We have identified "Cross-Domain Expansion" as a priority in Appendix A.2 (Future Works) and plan to collaborate with domain experts to extend our dataset to other scientific disciplines in the future.
>
> **We sincerely appreciate the reviewer's constructive suggestions and believe that the additional explanations significantly improve the quality of our submission. We hope that this provides sufficient reasons to raise the score.**
>
> ---
>
> [1] Landis JR, Koch GG. The measurement of observer agreement for categorical data. Biometrics. 1977 Mar;33(1):159-74. PMID: 843571.
>
> [2] Starace, Giulio, Oliver Jaffe, Dane Sherburn. PaperBench: Evaluating AI’s Ability to Replicate AI Research. arXiv:2504.01848, arXiv, 2025.
>
> [3] Du, Mingxuan, Benfeng Xu, Chiwei Zhu, Xiaorui Wang和Zhendong Mao. DeepResearch Bench: A Comprehensive Benchmark for Deep Research Agents. arXiv:2506.11763, arXiv, 2025..
>
> [4] Zheng, Lianmin, Wei-Lin Chiang, Ying Sheng, et al. Judging LLM-as-a-Judge with MT-Bench and Chatbot Arena. arXiv:2306.05685, arXiv, 2023.

---

> ### Author Response · Authors · 2025-11-28
> **Follow-up Comment: Clarification on Key Points & New Comparative Experiments**
>
> As the discussion period draws to a close, we would like to respectfully reiterate several key points regarding our methodology to ensure there are no lingering misunderstandings. Furthermore, we wish to highlight the **new comparative experiments with LLM-generated rubrics (Appendix G)** added in the revised paper, which also address reviewer’s concerns regarding "circular dependency" problem.
>
> To address specific concerns raised in reviewer’s previous comments, we wish to clarify:
>
> - **Rubrics are 100% Human-Designed**: The specific criteria and wording were fully authored by human experts. The LLM’s role was strictly limited to preliminary context extraction in Stage 1 to assist the human experts.
> - **Reliability of Automated Scoring**: Our judge model achieved an F1-score of 0.80 with human judgments, falling within the "Substantial" agreement range [1]. This result is comparable to concurrent works such as PaperBench (F1=0.83) [2,3.4], also confirming the reliability of our evaluation.
> - **Comprehensive Error Analysis**: We have added **Appendix E.6**, providing a granular breakdown of failure patterns (e.g., incompleteness vs. hallucination) across different system designs to enhance interpretability.
> - **Scope Specificity**: Our scope is always strictly **"Frontier AI Research”** throughout the paper (e.g., in the Title and Introduction), instead of "frontier research" in a general scientific sense.
> - **Nature of Assessment**: Rubrics are designed as guidelines to identify dimensions of insight rather than rigid checklists, and the factual assessment evaluates whether cited claims are supported by their own retrieved sources, rather than whether the model cites a specific "standard" bibliography.
>
>
> ### **New Experiments on Comment 6**
>
> To empirically address your concern in Comment 6 (that the benchmark might merely reflect "LLM-to-LLM imitation fidelity"), we conducted a new study in **Appendix G: "Human-Designed vs. LLM-Generated Rubrics: A Comparative Analysis"**.
>
> We compared our expert-designed rubrics against two LLM-generated baselines (including $\text{Rubric}_\text{ref}$, where Claude-3.7 generated criteria with the same reference materials). The results provide strong justification for the human-in-the-loop approach:
>
> - **LLM-generated rubrics** tended to be overly rigid (Recall=0.60) or focused heavily on "nice-to-have" details (Weight=1) rather than essential insights.
> - **Expert-designed rubrics** achieved better alignment with human evaluation and a more balanced weight distribution.
>
> This confirms that our rubrics capture nuanced expert reasoning that current LLMs cannot reliably generate on their own, mitigating the risk of "epistemic contamination."
>
> **We hope these clarifications and new empirical evidence adequately address reviewer’s concerns. We would be grateful if reviewer could consider these improvements in the final score.**
>
> ---
>
> [1] Landis JR, Koch GG. The measurement of observer agreement for categorical data. Biometrics. 1977 Mar;33(1):159-74. PMID: 843571.
>
> [2] Starace, Giulio, Oliver Jaffe, Dane Sherburn. PaperBench: Evaluating AI’s Ability to Replicate AI Research. arXiv:2504.01848, arXiv, 2025.
>
> [3] Du, Mingxuan, Benfeng Xu, Chiwei Zhu, Xiaorui Wang和Zhendong Mao. DeepResearch Bench: A Comprehensive Benchmark for Deep Research Agents. arXiv:2506.11763, arXiv, 2025..
>
> [4] Zheng, Lianmin, Wei-Lin Chiang, Ying Sheng, et al. Judging LLM-as-a-Judge with MT-Bench and Chatbot Arena. arXiv:2306.05685, arXiv, 2023.

---

### Author Response · Authors · 2025-11-30
**Summary Comment of Contributions and Rebuttal Updates**

We would like to provide a brief summary of our paper and the significant updates made during the rebuttal to assist Area Chairs' final decision.

1. *Core Contributions*

This paper introduces ResearcherBench, **the first benchmark specifically designed to evaluate Deep AI Research Systems (DARS) on frontier AI research questions**. Unlike existing benchmarks, we curate 65 frontier AI research questions from authentic scenarios (e.g., expert interviews, lab discussions) rather than synthetic queries, and propose a dual evaluation framework combining Expert-Designed Rubrics and Factual Assessment to measure insight quality and citation reliability.

2. *Key Rebuttal Updates & New Experiments*

During the rebuttal period, we conducted extensive new experiments to address reviewers' specific suggestions. These additions have significantly strengthened the paper's methodological rigor:

- **Appendix G: Human-Designed vs. LLM-Generated Rubrics**
  - *Addressed*: **Comment 2 of Reviewer oGgf (Score: 4)** asked for justification of the expert annotation cost, and **Comment 6 of Reviewer msmb (Score: 2)** raised concerns about "circular dependency" of LLM-generated rubrics.
  - *Result*: We compared our expert-designed rubrics against LLM-generated baselines. The results empirically prove that LLM-generated rubrics lack discriminative power (ceiling effects) and fail to identify essential criteria (weight=3). This validates that the human-in-the-loop methodology is indispensable for our benchmark.
  - *Note*: Reviewer oGgf explicitly stated *“I would be willing to increase my score once this concern has been addressed”*.

- **Appendix E.6: Error Analysis and Examples**
  - *Addressed*: **Comment 3 of Reviewer msmb (Score: 2)** requested granular analysis of failure modes.
  - *Result*: We provided a detailed breakdown of failure patterns (e.g., Incompleteness vs. Hallucination) across different system designs. This enhances the interpretability of our evaluation outcomes.

- **Appendix E.7: Pairwise Comparison Analysis**
  - *Addressed*: **Comment 1 of Reviewer Awa7 (Score: 8)** suggested incorporating pairwise comparisons.
  - *Result*: We implemented pairwise evaluation but discovered a severe length bias (correlation $r=0.9487$ between win rate and length). This negative result strongly validates our choice of Rubric Assessment as the more objective and reliable metric.

- **Updated Main Results with Confidence Intervals**
  - *Addressed*: **Follow-up Comment of Reviewer oGgf** requested statistical significance analysis.
  - *Result*: We updated the main result in Table 3 with **95% Bootstrap Confidence Intervals**. The analysis confirms that the performance gaps between top-tier DARS (e.g., OpenAI, Gemini) and baselines are statistically significant despite the limited size of the dataset.

3. *Response to Remaining Concerns*

Regarding **Reviewer msmb (Score: 2)**,  we have systematically addressed all points he mentioned below:

 - **Reliability & Authorship (Comments 1, 2, 4)**: We clarified that our rubrics are 100% human-authored and demonstrated that our Judge Model achieves Substantial Agreement (F1=0.80) with human experts, aligning with concurrent works.
 - **Granular Analysis (Comment 3)**: We added Appendix E.6 to provide the requested breakdown of failure modes.
 - **Scope Clarification (Comment 5)**: We clarified that our scope is strictly "frontier AI Research", not general scientific research.
 - **Methodological Misunderstanding (Comment 6)**: We clarified that our rubrics serve as guidelines to identify insight dimensions rather than rigid checklists, and our factual assessment validates support from retrieved sources rather than adherence to a fixed bibliography.

*Note*: Although we provided these comprehensive responses and new experiments, the reviewer msmb did not respond during the discussion period.

4. *Conclusion*

With strong support from Reviewers Awa7 and VdGR, and having actively addressed all the core concerns of Reviewer oGgf and msmb through extensive new experiments, ResearcherBench establishes a necessary standard for the emerging field of autonomous research agents. **We believe that these improvements address all major concerns, and we look forward to a positive outcome.**

---

### Meta-Review · Area_Chair_oE4E · 2026-01-05

**Summary:**

Based on the reviews and the authors’ rebuttal, I believe that several substantive concerns remain insufficiently addressed. As a result, I recommend rejecting this paper. Below, I summarize the main concerns raised by the reviewers and assess how the authors’ rebuttal responds to each of them:

Reviewer msmb:
1. Reliability of LLM-as-a-Judge: Automated scoring may be unreliable for nuanced research insights. Authors argue this level of agreement is comparable to prior benchmarks.
2. Circular dependency" of LLM-generated rubrics: The authors compared their expert-designed rubrics against LLM-generated baselines.
3. Lack of granular failure analysis: The authors added detailed error analysis (Appendix E.6) with failure modes and system-specific patterns.

Reviewer Awa7:
1. Pairwise comparison analysis: The authors implemented pairwise evaluation but discovered a severe length bias.

Reviewer oGgf:
1. Novelty of the task and evaluation.
2. Confidence intervals. The authors updated the results with confidence intervals.
3. Experimentally demonstrate that the proposed approach leads to improvements in LLM-based evaluation. The authors compared their expert-designed rubrics against LLM-generated baselines.

Reviewer VdGR:
1. Generalizability to other scientific domains
2. Interpretability

**Reviewer Concerns:**

Reviewer msmb: While the authors argue that their approach aligns with current community standards, the conceptual concern regarding the reliability of LLM-as-a-judge remains only partially addressed. Similarly, the concern about circular dependency is mitigated but not fully resolved.

Reviewer Awa7: This reviewer provided a relatively brief review and raised few concerns, all of which appear to have been addressed adequately in the rebuttal.

Reviewer oGgf: The concern regarding insufficient novelty does not appear to be fully addressed in the rebuttal.

Reviewer VdGR: The questions raised by VdGR are not critical and they are not very well demonstrated by empirical evidence from the authors.

**Reviewer Scores:**

Reviewer msmb: I anticipate that Reviewer msmb is unlikely to increase their score, or may increase it to 4 at most.

Reviewer Awa7: Although Reviewer Awa7 originally assigned a score of 8, the overall quality and depth of this review are comparatively lower.

Reviewer oGgf: I expect that Reviewer oGgf is unlikely to increase their score (remaining at 4).

Reviewer VdGR: I expect that Reviewer VdGR is unlikely to increase their score (remaining at 6).

Overall assessment: The reviews assigning higher scores appear to be of comparatively lower quality, while the more substantive concerns raised by higher-quality reviewers have not been adequately addressed by the authors.

---

### Decision · Program_Chairs · 2026-01-26

Reject